# LOW-PROBABILITY TOKENS SUSTAIN EXPLORATION IN REINFORCEMENT LEARNING WITH VERIFIABLE REWARD

## ABSTRACT

Reinforcement Learning with Verifiable Rewards (RLVR) has propelled Large Language Models in complex reasoning, yet its scalability is often hindered by a training bottleneck where performance plateaus as policy entropy collapses, signaling a loss of exploration. Previous methods typically address this by maintaining high policy entropy, yet the precise mechanisms that govern meaningful exploration have remained underexplored. Our analysis suggests that an unselective focus on entropy risks amplifying irrelevant tokens and destabilizing training. This paper investigates the exploration dynamics within RLVR and identifies a key phenomenon: the gradual elimination of what we term ***reasoning sparks***: a crucial subset of low-probability tokens such as "wait", that initiate diverse reasoning paths. We find that while abundant in pre-trained models, these sparks are systematically extinguished during RLVR due to over-penalization, leading to a degeneracy in exploration. To address this, we introduce Low-probability Regularization (Lp-Reg). Its core mechanism regularizes the policy towards a heuristic proxy distribution. This proxy is constructed by first applying a probability threshold to filter out noise tokens and then re-normalizing the distribution over the remaining candidates. This process effectively shields the exploratory tokens from destructive updates. Experiments show that Lp-Reg enables stable on-policy training for around 3,000 steps over 81,204 GPU-Hours, a regime where many baseline entropy-control methods collapse. This sustained exploration leads to state-of-the-art performance, achieving a 60.17% average accuracy on five math benchmarks, an improvement of 2.66% over prior methods.

## 1 INTRODUCTION

The advent of large reasoning models has reshaped the trajectory of artificial intelligence, with paradigmatic examples including OpenAI O1 (OpenAI et al., 2024) and DeepSeek-R1 (DeepSeek-AI et al., 2025). A central technique underpinning these systems is reinforcement learning with verifiable reward (RLVR), which assigns reward to verifiable solutions through rule-based verification. These models generate extended chain-of-thought (CoT) reasoning (Wei et al., 2023) to solve challenging problems in domains like mathematical olympiads (He et al., 2024b). However, a notable bottleneck emerges during RL training that limits its scalability, frequently culminating in a performance plateau and subsequent collapse. This failure is consistently accompanied by a rapid decay in policy entropy, indicating a severe loss of exploration capacity (Yu et al., 2025; Cui et al., 2025; Wang et al., 2025b).

Previous approaches have recognized this declining exploration, attempting to address it through various entropy control mechanisms. Methods such as adaptive entropy regularization (He et al., 2025), high entropy change blocking (Cui et al., 2025), or selective token updates (Wang et al., 2025b) aim to maintain higher entropy as a proxy for exploration. However, relying on overall entropy can be an indirect and imprecise tool. An indiscriminate focus on maximizing randomness risks amplifying noise and destabilizing training (Ömer Veysel Çağatan & Akgün, 2025), suggesting a deeper issue beyond simply the quantity of randomness.

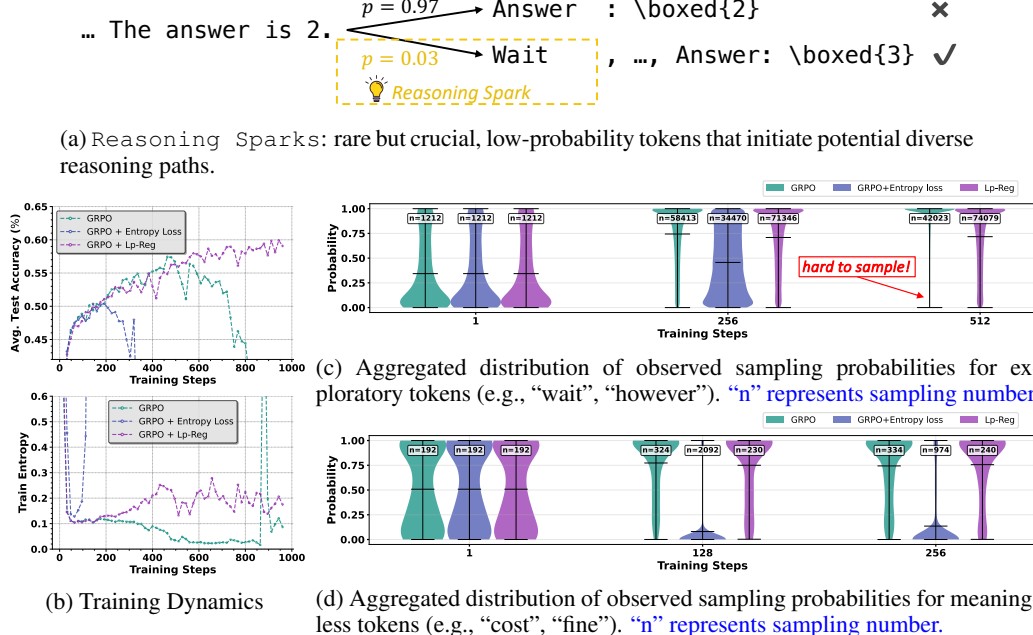

(a) `Reasoning Sparks`: rare but crucial, low-probability tokens that initiate potential diverse reasoning paths.

(c) Aggregated distribution of observed sampling probabilities for exploratory tokens (e.g., "wait", "however"). "n" represents sampling number.

(b) Training Dynamics

(d) Aggregated distribution of observed sampling probabilities for meaningless tokens (e.g., "cost", "fine"). "n" represents sampling number.

Figure 1: Selectively preserving low-probability tokens is key to overcoming performance plateaus in reasoning RL. **(a)** An example of *reasoning sparks*. **(b)** Standard GRPO training reaches a performance plateau and collapses, accompanied by decaying entropy. An indiscriminate entropy bonus (GRPO + Entropy Loss) leads to an even faster collapse. **(c)** We reveal the cause: GRPO systematically suppresses the low-probability sampling of entire classes of important *reasoning sparks*, causing their distributions to collapse towards high probabilities. Entropy loss fails to fix this. In contrast, our method, Lp-Reg, successfully preserves a healthy, wide distribution, sustaining exploration. **(d)** The failure of entropy bonuses is explained by their indiscriminate nature: they amplify the sampling of meaningless, low-probability noise, degrading exploration quality. The aggregated statistics in (c) and (d) demonstrate a systemic effect beyond single-token anecdotes. Detailed plots for individual tokens are available in Appendix C.4.

Our analysis suggests the performance bottleneck may stem from the systematic elimination of valuable low-probability exploratory tokens. We term these tokens ***reasoning sparks***; they include words like "wait", "however", or "perhaps", which often serve as logical connectives or expressions of uncertainty, naturally initiating diverse reasoning pathways (Figure 1a). As the aggregated violin plots in Figure 1c show, standard GRPO training systematically suppresses the low-probability sampling of these meaningful tokens. Furthermore, we find that indiscriminately boosting randomness amplifies meaningless noise—tokens such as "cost" or "fine" which are irrelevant to the mathematical reasoning context (Figure 1d). This amplification leads to an even faster performance collapse than the baseline, as shown in Figure 1b.

These findings present a central challenge: a successful exploration strategy should protect valuable, low-probability reasoning sparks without simultaneously amplifying the destructive effects of meaningless noise. To address this challenge, we introduce Low-probability Regularization (Lp-Reg). The primary goal of Lp-Reg is to preserve valuable low-probability tokens via regularization. To avoid amplifying noise, the method leverages a key observation: While both are empirically low-probability tokens, a meaningful exploratory token like "wait" often has a higher relative probability than a noise token like "cost" in the immediate next-token prediction. It is supported by the quantitative evidence in Section 6.3. Based on this insight, Lp-Reg first applies a probability threshold to filter out tokens treated as noise. It then re-normalizes the probability mass over the remaining candidates to construct a proxy distribution. In this proxy, the relative importance of valuable low-probability tokens is effectively increased. Finally, Lp-Reg penalizes the deviation of the original policy from this proxy using a forward KL divergence, which selectively protects the low-probability tokens that were preserved in the less-noisy proxy distribution.

Our experimental evaluation demonstrates the effectiveness of Lp-Reg. Our method enables stable on-policy training for around 3,000 steps over 81,204 GPU-hours, where many entropy-control methods have collapsed, resulting in better performance. On five widely used math benchmarks, this results in a 60.17% average accuracy on Qwen3-14B-Base, improving upon prior methods by at least 2.66%. Our contributions are summarized as follows:

- In contrast to prior work focusing on overall policy entropy, we identify the disappearance of *reasoning sparks* as a key issue and provide evidence that their preservation is crucial for sustained performance.

- We introduce Low-probability Regularization (Lp-Reg), a method that creates a more stable exploratory environment by filtering out presumed meaningless noise to protect the remaining low-probability tokens.

- We demonstrate through extensive experiments, utilizing a cumulative total of 300,000 GPU-hours, that Lp-Reg achieves state-of-the-art performance, while also enabling stable on-policy training over extended periods where baselines collapse.

- We provide a comprehensive analysis showing that our approach of filtering presumed meaningless noise yields superior results compared to entropy-control methods.

## 2 RELATED WORK

**Reinforcement learning for LLMs**   Recently, reinforcement learning has become the dominant framework for enhancing the reasoning abilities of large language models (LLMs) (OpenAI et al., 2024; DeepSeek-AI et al., 2025). By leveraging automatic checkers or symbolic verification, reinforcement learning with verifiable rewards (RLVR) achieved further breakthroughs in improving the reasoning capability of LLMs (Shao et al., 2024a; Yang et al., 2025a; Team et al., 2025). Based on RLVR and GRPO (Shao et al., 2024a), subsequent methods such as DAPO Yu et al. (2025), VAPO (Yue et al., 2025), and other policy optimization variants (Zhao et al., 2025; Cui et al., 2025; Zheng et al., 2025a) have been proposed to improve the stability, efficiency, and scalability of RL for reasoning models.

**Entropy collapse in RL training**   A recurring difficulty in training reasoning models with RL is the rapid collapse of policy entropy during the early stages of training. This phenomenon, which reflects excessive exploitation and insufficient exploration, has been widely recognized as a bottleneck for scaling RL in reasoning models. To mitigate collapse, researchers have explored several directions, including selectively regularizing updates at high-entropy "forking" tokens (Wang et al., 2025b), amplifying advantages at exploratory positions (Cheng et al., 2025), modifying clipping strategies (Yu et al., 2025; Zhao et al., 2025; Cui et al., 2025; Zheng et al., 2025a), or doing weight clipping (MiniMax et al., 2025; Su et al., 2025).

**Confidence-aware approaches**   An emerging line of work investigates how models' intrinsic confidence signals can guide exploration. Token probabilities naturally encode uncertainty and can indicate branching points in reasoning trajectories (Xu et al., 2025; Fu et al., 2025b; Hou et al., 2025; Wang et al., 2025a; Zheng et al., 2025b). Some works show that entropy minimization, which effectively sharpens the confidence of the model, can improve reasoning performance by encouraging the model to commit to consistent solution paths (Gao et al., 2025; Agarwal et al., 2025).

## 3 PRELIMINARIES

### 3.1 REINFORCEMENT LEARNING WITH VERIFIABLE REWARDS

Reinforcement learning (RL) has played a critical role in LLMs (Murphy, 2024). Formally,

$$\mathcal{J}_{\text{RL}}(\boldsymbol{\theta}) = \mathbb{E}_{(q,a)\sim D, o\sim\pi_\theta(\cdot|q)}\big[r(o, a)\big], \tag{1}$$

where $r(o, a)$ denotes the reward assigned to an output $o$ given a reference answer $a$. In reinforcement learning with verifiable rewards (RLVR), this reward is computed through rule-based

functions, such as Math-Verify[1]. Recent studies have demonstrated that large-scale RLVR encourages models to perform more deliberative reasoning by producing extended chains of thought prior to the final prediction, thereby substantially improving their capacity to solve complex problems (DeepSeek-AI et al., 2025). In practice, Eq. 1 is typically optimized using policy gradient methods, such as Proximal Policy Optimization (PPO) (Schulman et al., 2017) and Group Relative Policy Optimization (GRPO) (Shao et al., 2024b).

## 3.2 GROUP-RELATIVE POLICY OPTIMIZATION

GRPO is a representative actor-only policy gradient method for optimizing LLMs. It directly estimates the advantage of each token by leveraging multiple samples drawn from the same prompt. Formally, the advantage is defined as

$$A_{i,t} = \frac{R(o_i) - \text{mean}(\mathcal{G})}{\text{std}(\mathcal{G})},$$

(2)

where $\{o_1, \ldots, o_G\}$ are independent outputs sampled from the same prompt, with group size $G$, $\mathcal{G} = \{R(o_1), \ldots, R(o_G)\}$ denotes their associated rewards, and $R(o_i)$ is the reward of output $o_i$. In this formulation, $A_{i,t}$ represents the advantage of the $t$-th token in $o_i$. The policy is then optimized on the basis of these advantages using the PPO surrogate objective:

$$\mathcal{J}_{\text{GRPO}}(\boldsymbol{\theta}) = \mathbb{E}_{(q,a) \sim D, \{o_i\}_{i=1}^G \sim \pi_{\boldsymbol{\theta}_{\text{old}}}(\cdot|q)}$$

$$\frac{1}{\sum_{i=1}^G |o_i|} \sum_{i=1}^G \sum_{t=1}^{|o_i|} \Big[ \min \big[ r_{i,t} A_{i,t}, \, \text{clip}\big(r_{i,t}, 1-\epsilon, 1+\epsilon\big) A_{i,t} \big] - \beta \, D_{\text{KL}}\left(\pi_{\boldsymbol{\theta}} \, \| \, \pi_{\text{ref}}\right) \Big],$$

(3)

where $\beta$ controls the strength of KL regularization between the current policy $\pi_\theta$ and the reference policy $\pi_{\text{ref}}$. The probability ratio

$$r_{i,t} = \frac{\pi_\theta\big(o_{i,t} \mid q, o_{i,<t}\big)}{\pi_{\theta_{\text{old}}}\big(o_{i,t} \mid q, o_{i,<t}\big)}$$

(4)

serves as the importance sampling weight for off-policy training, where $\pi_{\theta_{\text{old}}}$ denotes the behavior policy. The hyperparameter $\epsilon$ specifies the clipping ratio, which constrains the updated policy from deviating excessively from the behavior policy, thereby ensuring stability during optimization.

## 4 LOW-PROBABILITY REGULARIZATION

To address the premature elimination of valuable *reasoning sparks*, we propose a regularization method termed **Low-probability Regularization (Lp-Reg)**. This method is designed to be integrated into policy gradient algorithms to create a more stable exploratory environment. The central idea is to leverage the model's own predictive distribution to construct a less-noisy reference for regularization, preserving low-probability tokens.

### 4.1 PROXY DISTRIBUTION $\pi_{\text{PROXY}}$

The foundation of Lp-Reg is the construction of a proxy distribution, which represents a filtered variant of the current policy $\pi_{\boldsymbol{\theta}}$. It is constructed in two steps:

1. **Filtering Noise Tokens:** We first define a set of noise tokens as those whose probability $\pi_{\boldsymbol{\theta}}(o|\cdot)$ under a threshold $\tau$. This threshold controls the filtering strategy, for which we explore two primary choices:

   - **Fixed threshold:** A simple approach where $\tau$ is a constant hyperparameter, e.g., 0.02.
   - **Min-p threshold:** Following (Nguyen et al., 2025), $\tau$ is defined relative to the peak probability: $\tau = \kappa \cdot \max_{o' \in V} \pi_{\boldsymbol{\theta}}(o'|\cdot)$, where $\kappa \in (0, 1)$ is a hyperparameter. This makes the filtering adaptive to the distribution's sharpness.

---

[1]`https://github.com/huggingface/Math-Verify`

Our primary experiments employ the min-p strategy for its adaptiveness, though fixed thresholds are also shown to be effective in our ablation studies.

2. **Probability Renormalization:** The proxy distribution $\pi_{\text{proxy}}$ assigns zero probability to tokens filtered out in the previous step and renormalizes the probability mass across the remaining tokens:

$$\pi_{\text{proxy}}(o|\cdot) = \begin{cases} \frac{\pi_{\boldsymbol{\theta}}(o|\cdot)}{\sum_{o' \text{ s.t. } \pi_{\boldsymbol{\theta}}(o'|\cdot)>\tau} \pi_{\boldsymbol{\theta}}(o'|\cdot)} & \text{if } \pi_{\boldsymbol{\theta}}(o|\cdot) > \tau \\ 0 & \text{otherwise} \end{cases}. \tag{5}$$

This process treats tokens with low relative probabilities as noise, while preserving all others.

### 4.2 Low-probability Regularization Objective

The Low-probability Regularization (Lp-Reg) penalty is integrated into the GRPO framework as a selective regularization term. The final objective function is:

$$
\begin{aligned}
J(\boldsymbol{\theta}) =& \mathbb{E}_{\mathcal{B}\sim\mathcal{D},(q,a)\sim\mathcal{B},\{o_i\}_{i=1}^G \sim \pi_{\boldsymbol{\theta}_{\text{old}}}(\cdot|q)} \left[ \frac{1}{\sum_{i=1}^G |o_i|} \sum_{i=1}^G \sum_{t=1}^{|o_i|} \left[ \text{clip}(r_{i,t}(\boldsymbol{\theta}),0,U) \cdot A_{i,t} \right.\right. \\
& - \beta \cdot \mathbb{I}\left[\pi_{\boldsymbol{\theta}}(o_{i,t}|q,o_{i,<t}) < \delta_\rho^{\mathcal{B}} \wedge \pi_{\text{proxy}}(o_{i,t}|q,o_{i,<t}) > 0 \wedge A_{i,t} < 0\right] \\
& \left.\left. \cdot \mathcal{D}_{\text{KL}}\left(\pi_{\text{proxy}}(\cdot|q,o_{i,<t}) \| \pi_{\boldsymbol{\theta}}(\cdot|q,o_{i,<t})\right) \right] \right]
\end{aligned}
\tag{6}
$$

The first term is the policy gradient objective from GRPO. We modify its clipping by removing the lower bound to avoid suppressing high-variance exploratory actions and adding a large upper bound $U$ for numerical stability.

The second term is the Lp-Reg penalty. It is activated by the indicator function $\mathbb{I}[\cdot]$ only for tokens that satisfy three conditions simultaneously: first, their sampling probability $\pi_{\boldsymbol{\theta}}$ is below a dynamic low-percentile threshold $\delta_\rho^{\mathcal{B}}$, which is calculated as the lowest $\rho$-th percentile of the sampling probabilities of all tokens within the current training batch $\mathcal{B}$; second, their probability in the proxy distribution $\pi_{\text{proxy}}$ is greater than zero; and third, the token receives a negative advantage signal ($A_{i,t} < 0$). This final condition ensures the regularization applies exclusively to tokens receiving a negative learning signal, preventing their potential over-penalization while leaving updates from positive experiences unaffected. Regarding two core hyperparameters $\kappa$ in $\tau = \kappa \cdot \max_{o' \in V} \pi_{\boldsymbol{\theta}}(o'|\cdot)$ and $\rho$ in $\delta_\rho^{\mathcal{B}}$, a data-driven guideline of value selection is provided in Section B.3 and a sensitivity analysis is presented in Appendix B.4 to assess its robustness.

We use the forward KL divergence, $\mathcal{D}_{\text{KL}}(\pi_{\text{proxy}}\|\pi_{\boldsymbol{\theta}})$ as the regularization function. It imposes a significant penalty when $\pi_{\boldsymbol{\theta}}(o|\cdot)$ approaches zero for a token $o$ with non-zero probability in $\pi_{\text{proxy}}$, providing a targeted penalty against token elimination without forcing the policy to strictly match the heuristic proxy distribution.

## 5 Experiments

### 5.1 Experimental Setup

**Baselines**  We compare Lp-Reg against a suite of strong baselines, including a foundational algorithm and several state-of-the-art methods designed to enhance exploration through entropy control. Our primary baseline is **GRPO** (Shao et al., 2024a), a value-free policy optimization algorithm that employs group-relative advantage estimation, making it a common choice for RLVR. To represent classical entropy regularization methods, we implement **GRPO + Entropy Loss**, which directly incorporates the principles of Maximum Entropy RL by adding a policy entropy bonus to the GRPO objective function. We also compare against several advanced methods: **Clip-Higher** (Yu et al., 2025), a core component of DAPO that encourages higher entropy by using an asymmetric clipping range in the PPO objective; **Selective High-Entropy Training (80/20)** (Wang et al., 2025b), a method that restricts policy gradient updates to only the top 20% of tokens with the highest generation entropy; **KL-Cov** (Cui et al., 2025), which prevents entropy collapse by applying a selective KL-divergence penalty to tokens with the highest covariance between their log probabilities and

advantages; and **GSPO** (Zheng et al., 2025a), which modifies the clipping mechanism to operate at the sequence level to promote higher training entropy.

**Training Settings** All experiments are conducted within the `veRL` (Sheng et al., 2024) framework to ensure a standardized and fair comparison. Our main comparisons are based on approximately 1,000 training steps for the Qwen3-14B-Base model and 800 for the Qwen2.5-32B model. Each training requires about $8,000$ GPU hours on 32 NVIDIA H20 GPUs for the 14B model and $16,000$ GPU hours on 64 NVIDIA H20 GPUs for the 32B model. To assess whether low-probability tokens sustain exploration in RLVR, we further trained the Qwen2.5-32B model for $3,000$ steps over $81,204$ GPU-hours with our Lp-Reg and evaluated its training stability.

For the reinforcement learning from verifier rewards (RLVR) phase, models are trained with a maximum response length of $8,192$ tokens. We use a global batch size of 256. For off-policy methods, we use a mini-batch size of 32, resulting in 8 gradient updates per rollout. It should be noted that in our experimental results, "step" refers to the rollout step for both on-policy and off-policy methods. To ensure a fair comparison, a "step" in our experimental results consistently refers to a single rollout for all methods. A constant learning rate of $1 \times 10^{-6}$ is applied without a warmup schedule. We set the group number as 8 for all GRPO-based methods. To ensure numerical stability, we set the policy gradient's clipping by setting the upper bound of the importance sampling ratio to $U = 10$. For our proposed Lp-Reg, which uses the min-p threshold, we set the probability percentile threshold $\rho$ to $0.5\%$ for Qwen2.5-32B, Qwen3-8B-Base, Llama3-OctoThinker-8B and $1\%$ for Qwen3-14B-Base, the KL regularization coefficient $\beta$ to $1.0$, and the min-p ratio $\kappa$ to $0.02$. The proxy distribution, $\pi_{\text{proxy}}$, is constructed from the data-generating policy ($\pi_{\theta_{\text{old}}}$ in the off-policy setting and the current policy $\pi_{\theta}$ in the on-policy setting). For all baseline methods, we adopt the hyperparameters specified in their original public implementations to ensure a faithful reproduction. Specifically for the GRPO + Entropy Loss baseline, we set the entropy coefficient to $0.002$ within the `verl` framework.

| Domains | Training datasets | Evaluation Benchmarks |
|---|---|---|
| Math | Dapo-Math-17K (Yu et al., 2025) | AIME24 (MAA), AIME25 (MAA), MATH-500 (Hendrycks et al., 2021), OlympiadBench (He et al., 2024a), Minerva Math (Lewkowycz et al., 2022) |
| Code | AReaL-boba-2-RL-Code Fu et al. (2025a) | LCB-v5, LCB-v6 Jain et al. (2024) |
| Science | SCP-116k Lu et al. (2025) | GPQA-diamond Rein et al. (2024) |

Table 1: Training datasets and evaluation benchmarks across various domains.

**Evaluation** For evaluation, we assess model performance across eight reasoning benchmarks in Table 1, spanning various domains including math, code, and science. For small benchmarks, we use sampled decoding with a temperature of $0.6$ to obtain a robust performance estimate, generating 16 independent responses per problem for AIME24 and AIME25, and 8 for GPQA-diamond, LCB-v5, and LCB-v6. For larger benchmarks like MATH-500, OlympiadBench, and Minerva, we utilize greedy decoding.

## 5.2 MAIN RESULTS

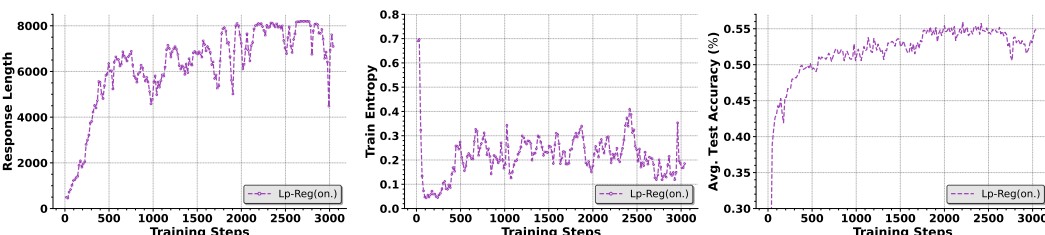

Figure 2: **Stable training over 3,000 training steps, totaling 81,204 GPU-hours**, for Lp-Reg (on-policy) on the Qwen2.5-32B-Base model.

As shown in Figure 2, Lp-Reg enables a stable reinforcement learning training for $3,000$ training steps, totaling $81,204$ GPU-hours for this single long-horizon run on the Qwen2.5-32B-Base model.

Furthermore, Figure 3 and Table 2 exhibit that Lp-Reg achieves state-of-the-art performance across five challenging mathematical reasoning benchmarks on both 14B and 32B model scales. On the Qwen3-14B model, on-policy Lp-Reg sets a new benchmark with an average accuracy of 60.17%, surpassing the next best method, 80/20, by 2.66%. Notably, Lp-Reg's advantage is more pronounced on the newer Qwen3-14B base model compared to the older Qwen2.5-32B. We hypothesize that as base models improve, their capacity for nuanced, low-probability reasoning increases, creating richer *reasoning sparks* for Lp-Reg to leverage, thereby amplifying its effectiveness. Note that the scores reported here correspond to the single checkpoint achieving the highest average accuracy. As aggregated metrics can sometimes obscure the model's peak potential on individual tasks, we provide a detailed analysis of per-benchmark peak scores in Appendix B.5.

Our experiments consistently show the superiority of on-policy training over off-policy methods across 14B and 32B scales. This is due to the inherent stability of on-policy updates, which avoid distribution shifts caused by mismatched data-sampling and training policies. Off-policy methods, such as Clip-Higher, often rely on importance sampling clipping, leading to instability. While competitive on Qwen2.5-32B, Clip-Higher's performance drops on Qwen3-14B, highlighting its fragility. In contrast, Lp-Reg's self-contained, policy-intrinsic regularization ensures its effectiveness in both on-policy and off-policy settings, unlike competing methods that are heavily reliant on off-policy importance sampling.

Beyond raw performance, Lp-Reg demonstrates a distinct entropy signature indicative of a healthy exploration-exploitation balance. As shown in Figure 3, methods like Clip-Higher induce a continuous, often artificial increase in policy entropy. Lp-Reg, however, facilitates a dynamic, multi-phase entropy trajectory: entropy initially decreases as the model learns core reasoning patterns, then gradually increases to foster exploration as performance improves, and finally stabilizes within a healthy range as accuracy converges. This adaptive behavior stems from confidence-aware regularization, which selectively protects valuable *reasoning sparks* without amplifying indiscriminate, high-entropy noise.

To further validate the generalizability of Lp-Reg, we extend the experimental comparison to additional domains (e.g. science, code), as well as various model sizes (e.g., 8B) and architectures (e.g., Llama). Details can be found in Appendix B.6.

| Method | AIME24 | AIME25 | Math-500 | Minerva | Olympiad Bench | Avg. |
|---|---|---|---|---|---|---|
| **Qwen2.5-32B-Base** (800 training steps) | | | | | | |
| GRPO (Shao et al., 2024a) (off.) | 30.63 | 22.29 | 88.00 | 41.18 | 54.37 | 47.29 |
| GSPO (Zheng et al., 2025a) (off.) | 33.33 | 22.29 | 87.60 | **48.53** | 55.56 | 49.46 |
| Clip-Higher (Yu et al., 2025) (off.) | **38.33** | **29.79** | 87.60 | 45.22 | 56.44 | 51.48 |
| KL-Cov (Cui et al., 2025) (off.) | 35.62 | 27.50 | 87.40 | 44.49 | 55.11 | 50.02 |
| 80/20 (Wang et al., 2025b) (off.) | 38.12 | 28.75 | 87.00 | 45.22 | 58.37 | 51.49 |
| Lp-Reg (off.) | 37.71 | 24.58 | **90.20** | 40.81 | 59.70 | 50.60 |
| GRPO (Shao et al., 2024a) (on.) | 28.54 | 22.50 | 86.60 | 44.85 | 60.30 | 48.56 |
| GRPO + Entropy Loss (on.) | 3.75 | 1.88 | 60.80 | 27.94 | 22.22 | 23.32 |
| 80/20 (Wang et al., 2025b)(on.) | 32.50 | 28.54 | 89.40 | 45.59 | 57.63 | 50.73 |
| Lp-Reg (on.) | 38.12 | 27.08 | 90.00 | 46.32 | **61.19** | **52.54** |
| **Qwen3-14B-Base** (1,000 training steps) | | | | | | |
| GRPO (Shao et al., 2024a) (off.) | 34.38 | 27.08 | 89.20 | 49.26 | 55.70 | 51.13 |
| GSPO (Zheng et al., 2025a) (off.) | 41.46 | 34.58 | 88.60 | **50.74** | 59.85 | 55.05 |
| Clip-Higher (Yu et al., 2025) (off.) | 41.67 | 32.71 | **95.00** | 47.43 | 64.00 | 56.16 |
| KL-Cov (Cui et al., 2025) (off.) | 49.17 | 34.79 | 93.00 | 47.43 | 62.07 | 57.29 |
| 80/20 (Wang et al., 2025b) (off.) | 43.96 | 34.58 | 91.80 | 48.16 | 60.89 | 55.88 |
| Lp-Reg (off.) | 46.25 | 34.17 | 92.40 | 48.16 | 64.44 | 57.08 |
| GRPO (Shao et al., 2024a) (on.) | 46.04 | 34.38 | 93.00 | 48.53 | 65.19 | 57.43 |
| GRPO + Entropy Loss (on.) | 37.29 | 25.21 | 88.20 | 46.32 | 54.96 | 50.40 |
| 80/20 (Wang et al., 2025b) (on.) | 47.29 | 32.50 | 91.60 | 50.37 | 65.78 | 57.51 |
| Lp-Reg (on.) | **50.83** | **37.92** | 94.40 | 49.26 | **68.44** | **60.17** |

Table 2: Main results on five mathematical reasoning benchmarks. On-policy (on.) and off-policy (off.) training methods are highlighted with distinct colors. For each method, all reported scores are derived from the single checkpoint that achieved the highest average accuracy across the five benchmarks. Best scores are **bolded** while second-best scores are underlined.

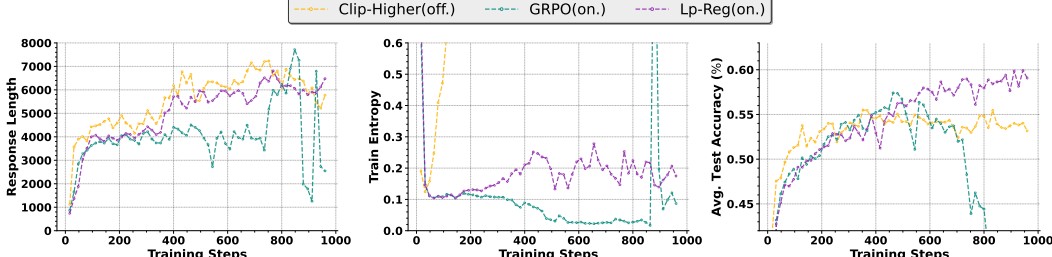

Figure 3: Training dynamics on the Qwen3-14B-Base model. To best illustrate the performance differences, we compare the top-performing methods. Lp-Reg demonstrates superior and stable performance. Full training dynamics for the Qwen2.5-32B model are available in Appendix B.1.

## 5.3 ABLATION STUDY

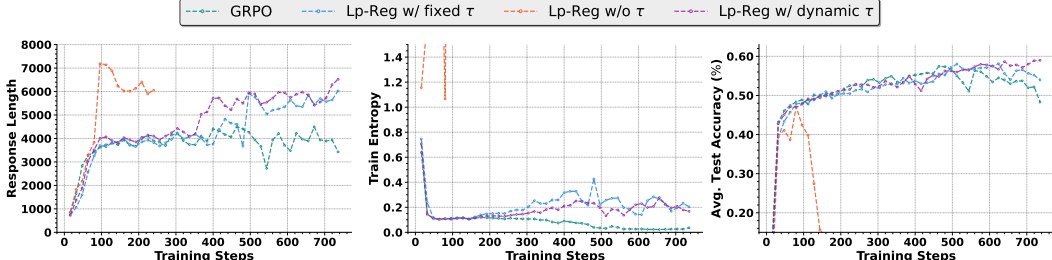

Figure 4: Ablation studies for core components of Lp-Reg on the Qwen3-14B-Base model. The results confirm that targeting our noise filtering threshold $\tau$ is critical for stable performance. The adaptiveness of the min-p threshold is also shown to be beneficial over a fixed one.

We conduct a series of ablation studies to dissect the core components of Lp-Reg and validate our key design choices.

**Importance of Noise Filtering.** Lp-Reg only protects tokens deemed meaningful by the proxy distribution ($\pi_{\text{proxy}} > 0$). To test this, we remove the filter and fork all tokens below the noise threshold $\tau$ from contributing to gradient updating (Lp-Reg w/o $\tau$). Figure 4 shows that this leads to a catastrophic performance collapse and entropy explosion. This confirms that filtering is critical to ignore the extreme tail of the distribution, which consists of incoherent noise that destabilizes training if regularized.

**Dynamic vs. Fixed Threshold.** We conduct a comparison between the dynamic min-p noise threshold (Lp-Reg w/ dynamic $\tau$) and the fixed noise threshold (Lp-Reg w/ fixed $\tau$) in Section 4.1. As shown in Figure 4, the fixed threshold underperforms compared to the dynamic threshold, which we adopt as the default. However, it still significantly surpasses the standard GRPO. This indicates that while the core filtering principle is effective, the dynamic nature of min-p provides a more robust estimate of the model's confidence across different contexts, better preserving genuine *reasoning sparks*.

We conduct further ablation studies on the high-entropy token regularization. For detailed results and analysis, please refer to Appendix B.2.

## 6 ANALYSIS

To understand the mechanisms behind Lp-Reg's performance, we conduct a series of analyses focusing on how it overcomes the exploration bottleneck by targeting and preserving valuable reasoning tokens.

### 6.1 PROBABILITY-ENTROPY DISTRIBUTION OF REASONING SPARKS

We begin by exploring the distinction between low-probability tokens and high-entropy tokens. Figure 5 highlights this contrast by comparing tokens from the top 1% lowest probability with those from the top 1% highest entropy. The difference is striking: low-probability tokens frequently include semantically meaningful exploratory markers such as *"But," "Wait,"* and *"Perhaps,"* which often signal a shift in the reasoning trajectory. In contrast, high-entropy tokens are dominated by ubiquitous function words (e.g., *"the," "of"*) or formatting symbols (e.g., `frac`), which carry little exploratory intent. This explains why entropy-based regularization often fails to enhance exploration: it confuses noise with exploration.

However, the set of low-probability tokens is also not uniformly useful. It also includes noisy artifacts such as spurious newline characters (`\n`) or formatting debris, whose regularization can destabilize training rather than enhance reasoning. To mitigate this, Lp-Reg applies a threshold $\tau$ that filters out such noise. Ablation studies in Section 5.3 confirm the necessity of this step: removing the threshold results in unstable training dynamics and degraded reasoning performance. Thus, Lp-Reg's effectiveness stems not only from targeting low-probability tokens but also from selectively excluding meaningless tokens.

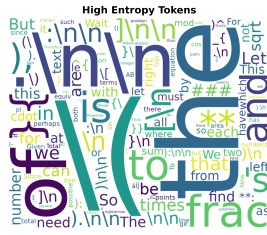
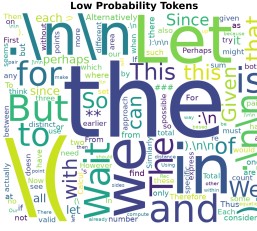

Figure 5: The word cloud statistics.

## 6.2 TOKEN PROBABILITY DYNAMICS

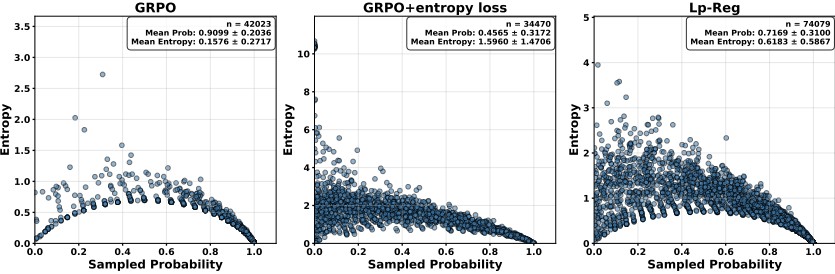

Figure 6: Probability–Entropy scatter plots of explorative tokens.

Figure 6 shows the probability–entropy distributions of key explorative tokens ("but", "wait", "perhaps", "alternatively", and "however") under three methods: GRPO, GRPO + Entropy Loss, and our Lp-Reg.

With the baseline GRPO, these tokens are concentrated in low-entropy, high-probability regions. In this case, tokens like "wait" tend to appear only when the model is already confident, turning them into deterministic patterns rather than initiating a new exploration path with uncertainty.

Adding an entropy loss changes this behavior, but in an uncontrolled way. Some sampled "wait" tokens appear at extremely high entropy levels (sometimes exceeding 10), which superficially boosts diversity but produces little useful exploratory signal. These scattered occurrences do not integrate meaningfully into the reasoning process.

Our Lp-Reg method yields a more balanced dynamic. Explorative tokens are sampled across a broad range of entropy values, from high probability to low probability states. This balance prevents their probabilities from collapsing under negative feedback while keeping them informative for reasoning. As a result, tokens like "wait" remain viable options throughout training, allowing the model to explore alternative reasoning paths rather than overfitting to fixed usage patterns.

Figure 7 further compares the frequency of explorative tokens ("but", "wait", "perhaps", "alternatively", and "however") under GRPO and Lp-Reg. Our method consistently maintains a higher fraction of these tokens, demonstrating that Lp-Reg not only broadens their probability–entropy distribution but also sustains their practical use throughout training.

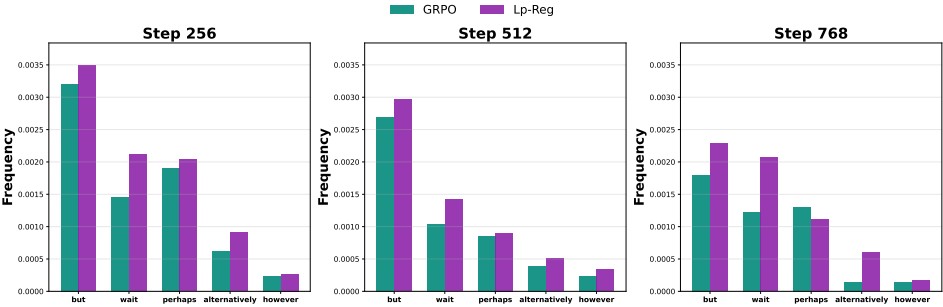

Figure 7: Frequency of explorative tokens during training.

### 6.3 PROBABILISTIC DISTINCTION BETWEEN REASONING SPARKS AND NOISE

Our introduction established a challenge for a successful exploration strategy: it must protect valuable, low-probability *reasoning sparks* without simultaneously amplifying the destructive effects of irrelevant noise. This raises a critical question: is there a systemic, observable difference between these two classes of tokens within the low-probability range that our method can exploit?

To investigate this, we analyze the next-token prediction distribution throughout the training process. Due to storage limitations, we focus our analysis on the top-64 most probable tokens, but specifically examine those within a low-probability range (0 to 0.1) to isolate the phenomenon from high-probability tokens. Figure 8 plots the average probability of two distinct classes of tokens over time: a group of meaningful exploratory tokens (e.g., "wait", "perhaps") and a group of irrelevant tokens (e.g., "cost", "fine").

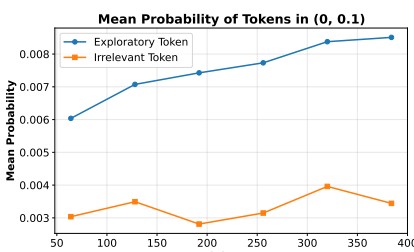

Figure 8: Probabilistic distinction between exploratory and irrelevant tokens across training steps in standard GRPO training.

The results reveal a clear and consistent statistical distinction: across all training stages, the average next-token probability of meaningful exploratory tokens is persistently higher than that of irrelevant tokens. It can be attributed to the intrinsic confidence of LLMs (Nguyen et al., 2025; Xu et al., 2025; Fu et al., 2025b). This persistent probabilistic gap provides the foundational justification for our Lp-Reg design. It suggests that while a perfect separation is not possible, a probability threshold $\tau$, as defined for our proxy distribution in Section 4.1, can serve as a principled filtering mechanism. By setting such a threshold, we can effectively filter out a substantial portion of the lowest-probability irrelevant tokens, which constitute destabilizing noise, while simultaneously retaining a majority of the valuable exploratory tokens that give rise to *reasoning sparks*. This allows Lp-Reg to focus its regularization on tokens that are more likely to be meaningful, providing a targeted and robust approach to preserving high-quality exploration.

## 7 CONCLUSION

In this work, we investigated the exploration collapse in Reinforcement Learning with Verifiable Rewards, identifying a key mechanism driving this failure: the systematic elimination of valuable, low-probability *reasoning sparks*. To address this, we introduced Low-probability Regularization (Lp-Reg), a method designed to selectively preserve these crucial exploratory pathways. Lp-Reg leverages the insight that *reasoning sparks* often exhibit higher relative probabilities than meaningless noise in their immediate predictive context. By filtering out the noise tokens and regularizing the policy towards the remainder, our method effectively protects valuable sparks from being extinguished. This focus on exploration quality over quantity enables stable on-policy training for around 3,000 steps, resulting in at least 2.66% test accuracy improvement over baselines and underscoring the importance of preserving the policy's useful low-probability tail.

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

# Appendix

## A   LARGE LANGUAGE MODELS USAGE STATEMENT

In adherence to the ICLR 2026 policy, we disclose the use of a large language model (LLM) as a general-purpose writing assistant during the preparation of this manuscript. The LLM's role was strictly limited to improving the clarity, grammar, and readability of our author-written text, such as spell-checking and rephrasing sentences for better flow. Crucially, the LLM did not contribute to any of the core scientific aspects of this work, including research ideation, experimental design, data analysis, or the generation of novel insights. The authors have carefully reviewed all LLM-modified text and take full responsibility for the intellectual substance and final content of this paper.

## B   DETAILS OF EXPERIMENTS

### B.1   TRAINING DYNAMICS ON QWEN2.5-32B

The training dynamics of Lp-Reg and other state-of-the-art RLVR methods on the Qwen2.5-32B base model are presented in Figure 9. The results show that Lp-Reg maintains a comparable performance in test accuracy throughout the training process, underscoring the benefits of our low-probability token regularization strategy for preventing exploration collapse.

### B.2   FURTHER ABLATION STUDY

To verify that targeting low-probability tokens is superior to the conventional wisdom of targeting high entropy, we conduct a comparison between the high-entropy token regularization (highest $\mathcal{H}$)

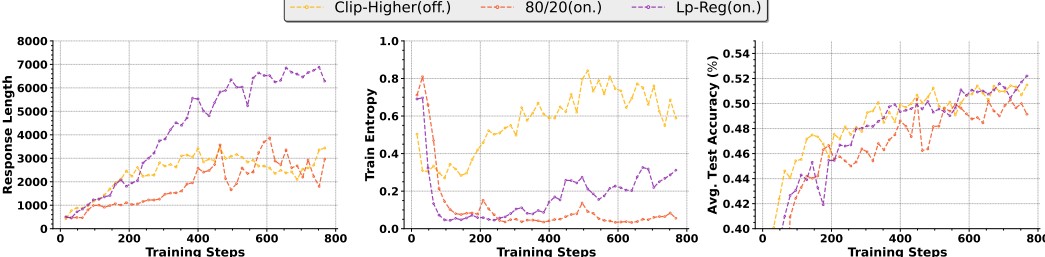

Figure 9: Training dynamics on the Qwen2.5-32B-Base model. To best illustrate the performance differences, we compare the top-performing methods. Lp-Reg demonstrates superior and more stable performance throughout training.

and the low-probability regularization (lowest $\pi_\theta$, vanilla Lp-Reg). Instead of applying Lp-Reg to the lowest 1% probability tokens, we apply an identical regularization mechanism to the tokens with the highest 1% entropy. As shown in Figure 10, this approach not only fails to improve performance but also fails to sustain high entropy, which collapses after an initial spike. This result reinforces our claim from the Introduction: high entropy is a poor proxy for valuable exploration. As our analysis in Section 6.1 further corroborates, high-entropy tokens are often common function words or formatting characters, not true *Reasoning Sparks*. Regularizing them pollutes the learning signal without protecting the structured, low-probability reasoning paths necessary for progress.

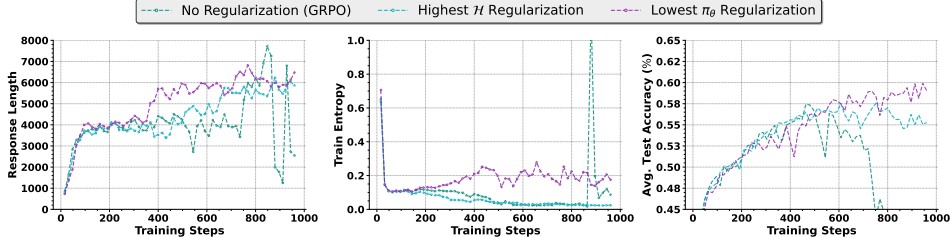

Figure 10: Ablation study comparing low-probability token regularization versus high-entropy token regularization for Lp-Reg (on-policy) on the Qwen3-14B-Base model.

## B.3 GUIDELINES FOR HYPERPARAMETER SELECTION

In this section, we provide a data-driven guideline for selecting the initial values of the two core hyperparameters in Lp-Reg: the low-probability percentile $\rho$ and the min-p ratio $\kappa$. Here, $\rho$ determines the regularization threshold $\delta_\rho^{\mathcal{B}}$, while $\kappa$ defines the noise filtering threshold $\tau = \kappa \cdot \max_{o' \in V} \pi_\theta(o'|\cdot)$. Instead of heuristic guessing, we derive the rational ranges for these parameters by analyzing the training dynamics of the standard GRPO baseline.

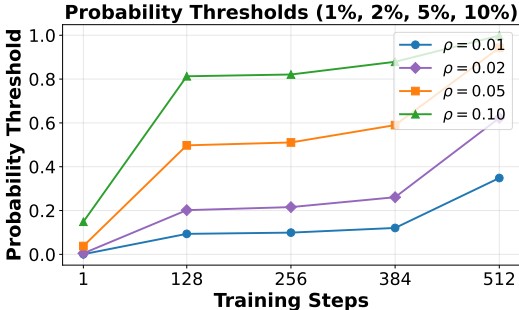

Figure 11: Evolution of probability thresholds for different percentiles ($\rho$) during standard GRPO training. The bottom 1% ($\rho = 0.01$) consistently captures the low-probability tail ($< 0.1$), whereas higher percentiles include high-confidence tokens.

**Selection of** $\rho$**.** Figure 11 visualizes the upper probability bound of tokens falling within the lowest $\rho$ percentile during standard GRPO training. As illustrated, RLVR training causes the policy

distribution to collapse, concentrating mass on high-probability tokens. From step 128 to 384, the probability of tokens in the bottom $1\%$ consistently remains in the strictly low-probability regime ($< 0.1$). In contrast, tokens in the bottom $5\%$ span a much wider range, reaching probabilities as high as $0.5$, which are no longer low-probability candidates requiring protection. Consequently, setting $\rho \approx 1\%$ ($0.01$) is a logical and robust choice to target the true tail of the distribution without inadvertently regularizing high-probability tokens. The sensitivity analysis in Figure 13 confirms that performance is stable around this empirically derived value.

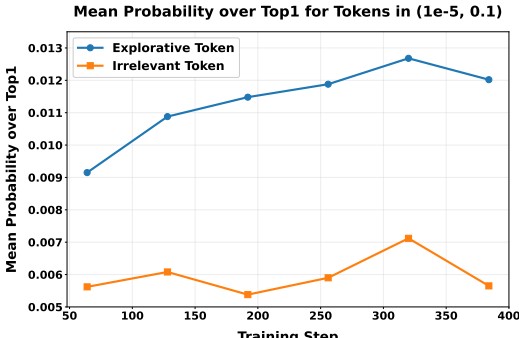

Figure 12: Comparison of relative probability ratios between exploratory tokens and meaningless noise tokens during training. A clear gap exists, supporting the selection of $\kappa \approx 0.01$.

**Selection of $\kappa$.** Figure 12 compares the average relative probability ratio ($\frac{\pi_{\boldsymbol{\theta}}(o|\cdot)}{\max_{o' \in V} \pi_{\boldsymbol{\theta}}(o'|\cdot)}$) between a set of meaningful exploratory tokens ($S_{explore} = \{$"but", "wait", "perhaps", "alternatively", "however"$\}$) and a set of meaningless noise tokens ($S_{noise} = \{$"cost", "fine", "balanced", "ere", "trans"$\}$) that are irrelevant with the reasoning task. The statistics, derived from standard GRPO training, reveal a distinct and persistent separability gap: the relative probability of meaningful exploratory tokens consistently exceeds that of noise tokens throughout the training process. This empirical gap justifies setting the min-p ratio $\kappa$ (which determines the noise threshold $\tau = \kappa \cdot \max_{o' \in V} \pi_{\boldsymbol{\theta}}(o'|\cdot)$) within this separation region. As shown in the figure, most noise tokens typically fall below a ratio of $0.01$, while exploratory tokens remain above it. Therefore, values of $\kappa$ around $0.01$ (or slightly higher) serve as effective initial settings to filter noise while preserving reasoning sparks. The robustness of Lp-Reg with $\kappa \in \{0.01, 0.02, 0.03\}$, as verified in Section B.4, further validates this selection strategy.

### B.4 HYPERPARAMETER SENSITIVITY ANALYSIS

In this section, we analyze the sensitivity of two core hyperparameters in Lp-Reg to demonstrate the robustness of our method: the low-probability percentile $\rho$ and the min-p ratio $\kappa$. The results are presented in Figure 13.

The parameter $\rho$, as defined in our objective function (Equation 6), determines the percentile threshold for identifying low-probability tokens that are candidates for regularization. A higher $\rho$ means a wider range of tokens are protected. As shown in the top panel of Figure 13, we evaluated $\rho$ with values of $0.005$, $0.010$, and $0.015$. The training trajectories for average test accuracy are nearly identical, and the final performance across all three settings is highly comparable. This indicates that Lp-Reg is not overly sensitive to the precise scope of tokens being protected within this reasonable range.

The hyperparameter $\kappa$ controls the adaptiveness of the min-p filtering threshold, which defines the boundary for what is treated as noise. A smaller $\kappa$ results in a more conservative filtering strategy, removing fewer tokens. Our sensitivity analysis for $\kappa$, presented in the bottom panel of Figure 13, shows a similar trend of stability. Across the tested values of $0.01$, $0.02$, and $0.03$, the training curves and final performance remain consistently high and tightly clustered. Taken together, these results demonstrate the robustness of Lp-Reg. The method's effectiveness is not contingent on extensive, fine-grained hyperparameter tuning, highlighting its practical applicability.

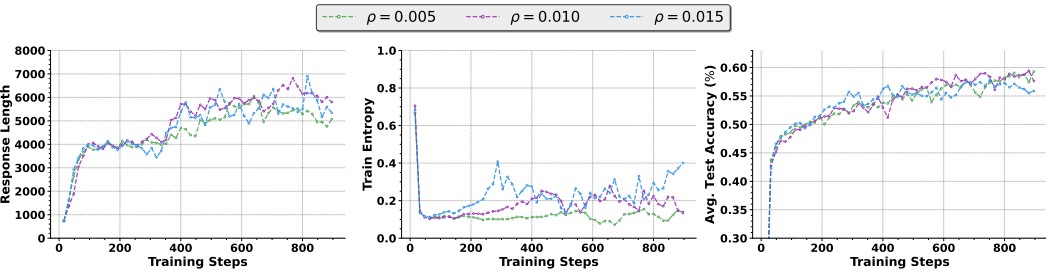

(a) Sensitivity analysis on $\rho$ which defined the low-probability percentile threshold $\delta_\rho^{\mathcal{B}}$.

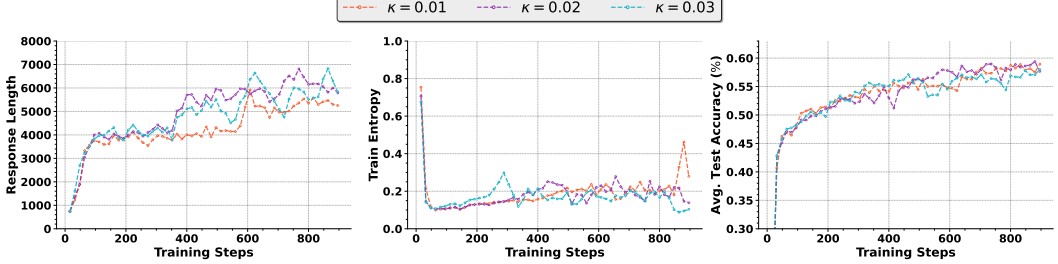

(b) Sensitivity analysis on $\kappa$ which defined the noise threshold $\tau = \kappa \cdot \max_{o' \in V} \pi_{\boldsymbol{\theta}}(o'|\cdot)$

Figure 13: Training dynamics of Lp-Reg method with different hyperparameters.

| Methods | AIME24 | AIME25 | Math-500 | Minerva | Olympiad Bench |
|---|---|---|---|---|---|
| **Qwen2.5-32B-Base** | | | | | |
| GRPO (Shao et al., 2024a) (off.) | 30.63 | 23.75 | 88.00 | 46.69 | 56.00 |
| GSPO (Zheng et al., 2025a) (off.) | 36.88 | 26.46 | 89.00 | **49.63** | 56.30 |
| Clip-Higher (Yu et al., 2025) (off.) | 39.58 | **32.71** | 88.80 | 48.90 | 58.22 |
| KL-Cov (Cui et al., 2025) (off.) | 36.88 | 29.38 | 89.00 | 48.16 | 56.89 |
| 80/20 (Wang et al., 2025b) (off.) | 40.62 | 30.21 | 90.80 | 48.16 | 58.81 |
| Lp-Reg (off.) | 37.71 | 26.88 | 90.20 | 43.38 | 60.15 |
| GRPO (Shao et al., 2024a) (on.) | 32.50 | 23.54 | 88.80 | 47.79 | 60.30 |
| GRPO + Entropy Loss (on.) | 3.75 | 2.50 | 60.80 | 32.72 | 22.22 |
| 80/20 (Wang et al., 2025b) (on.) | 35.00 | 28.54 | 90.00 | 47.79 | 58.81 |
| Lp-Reg (on.) | **45.00**$_{+10.78\%}$ | **32.71**$_{+0.00\%}$ | **93.00**$_{+2.42\%}$ | 48.16$_{-2.96\%}$ | **64.15**$_{+6.38\%}$ |
| **Qwen3-14B-Base** | | | | | |
| GRPO (Shao et al., 2024a) (off.) | 35.83 | 27.71 | 91.00 | 48.16 | 59.56 |
| GSPO (Zheng et al., 2025a) (off.) | 43.75 | 36.67 | 91.60 | 50.74 | 61.04 |
| Clip-Higher (Yu et al., 2025) (off.) | 44.79 | 33.75 | **95.00** | 49.63 | 65.19 |
| KL-Cov (Cui et al., 2025) (off.) | 49.38 | 35.83 | 94.20 | **51.84** | 64.44 |
| 80/20 (Wang et al., 2025b) (off.) | 44.17 | 34.58 | 92.80 | 50.37 | 62.81 |
| Lp-Reg (off.) | 48.75 | 34.79 | 94.40 | 49.63 | 65.78 |
| GRPO (Shao et al., 2024a) (on.) | 46.04 | 35.42 | 93.80 | 50.37 | 65.63 |
| GRPO + Entropy Loss (on.) | 37.29 | 28.54 | 90.60 | 48.53 | 57.93 |
| 80/20 (Wang et al., 2025b) (on.) | 47.29 | 35.00 | 94.00 | 50.37 | 65.78 |
| Lp-Reg (on.) | **51.88**$_{+5.06\%}$ | **40.62**$_{+10.77\%}$ | **95.00**$_{+0.00\%}$ | 51.47$_{-0.71\%}$ | **70.37**$_{+6.98\%}$ |

Table 3: Per-benchmark peak performance on five mathematical reasoning benchmarks. **Note that the scores reported represent the maximum value achieved for each specific benchmark individually; thus, scores within a single row may originate from different training checkpoints.** Best scores are **bolded** while second-best scores are underlined. The relative accuracy improvement of Lp-Reg over the next best method is indicated as a subscript.

## B.5 PER-BENCHMARK PEAK PERFORMANCE ANALYSIS

In Section 5.2, we reported performance based on a single checkpoint selected for the best average test accuracy across five mathematical benchmarks. However, aggregating results can obscure the model's peak potential on individual tasks. To address this, we present the per-benchmark

best scores in Table 3. As shown, our on-policy Lp-Reg achieves the highest peak scores on all benchmarks with the exception of Minerva. Even on Minerva, where Lp-Reg(on.) is not the best performer, the gap is marginal: on Qwen2.5-32B-Base, it trails the highest score by only $1.47$ percentage points (a relative difference of $-2.96\%$). Conversely, the gains on other benchmarks are substantial, particularly on the most challenging reasoning tasks such as AIME24, AIME25, and Olympiad Bench. Notably, on Qwen2.5-32B-Base, Lp-Reg(on.) outperforms the second-best method, 80/20(off.), by a relative margin of $10.78\%$ on AIME24. Similarly, on Qwen3-14B-Base, it achieves a $10.77\%$ relative improvement on AIME25. These significant improvements on the hardest benchmarks underscore the effectiveness of Lp-Reg in solving complex reasoning problems.

We further evaluate the exploration capability of our method by comparing the best pass@k rates. As detailed in Table 4, Lp-Reg(on.) consistently achieves the highest pass@k scores on both AIME24 and AIME25 across both model scales, often by a wide margin. For the Qwen2.5-32B model, Lp-Reg(on.) demonstrates a minimum relative improvement of $5.97\%$ in pass@k metrics on AIME24. Furthermore, on the Qwen3-14B model, it shows impressive gains on AIME25, achieving relative improvements ranging from $7.81\%$ to $9.33\%$. These robust pass@k results provide strong evidence that Lp-Reg effectively sustains meaningful exploration throughout long-horizon RLVR training, resulting in more diverse and successful reasoning rollouts.

| Methods | AIME24 | | | AIME25 | | |
|---|---|---|---|---|---|---|
| | Pass@2 | Pass@4 | Pass@8 | Pass@2 | Pass@4 | Pass@8 |
| **Qwen2.5-32B-Base** | | | | | | |
| GRPO (Shao et al., 2024a) (off.) | 40.06 | 49.87 | 58.10 | 29.11 | 36.25 | 44.75 |
| GSPO (Zheng et al., 2025a) (off.) | 46.83 | 57.62 | 66.78 | 32.86 | 38.84 | 45.04 |
| Clip-Higher (Yu et al., 2025) (off.) | 48.11 | 57.80 | 68.32 | 35.92 | 43.27 | 51.29 |
| KL-Cov (Cui et al., 2025) (off.) | 46.89 | 55.94 | 64.61 | 35.44 | 41.60 | 49.39 |
| 80/20 (Wang et al., 2025b) (off.) | 48.97 | 56.52 | 64.29 | 34.08 | 41.35 | 49.47 |
| Lp-Reg (off.) | 49.69 | 59.75 | 69.21 | 33.75 | 42.44 | 50.80 |
| GRPO (Shao et al., 2024a) (on.) | 42.08 | 51.74 | 61.95 | 29.19 | 35.83 | 43.20 |
| GRPO + Entropy Loss (on.) | 6.89 | 11.88 | 19.08 | 4.00 | 6.06 | 10.11 |
| 80/20 (Wang et al., 2025b) (on.) | 45.06 | 55.33 | 63.40 | 35.28 | 41.64 | 48.54 |
| Lp-Reg (on.) | **53.33**$_{+7.33\%}$ | **63.50**$_{+6.28\%}$ | **73.34**$_{+5.97\%}$ | **38.28**$_{+6.57\%}$ | **45.52**$_{+5.20\%}$ | **53.12**$_{+3.57\%}$ |
| **Qwen3-14B-Base** | | | | | | |
| GRPO (Shao et al., 2024a) (off.) | 45.31 | 54.81 | 64.09 | 34.14 | 41.00 | 48.29 |
| GSPO (Zheng et al., 2025a) (off.) | 54.11 | 63.67 | 71.05 | 44.39 | 51.97 | 59.67 |
| Clip-Higher (Yu et al., 2025) (off.) | 56.00 | 66.85 | 74.91 | 40.19 | 48.31 | 57.35 |
| KL-Cov (Cui et al., 2025) (off.) | 59.47 | 66.84 | 74.52 | 42.22 | 49.98 | 58.65 |
| 80/20 (Wang et al., 2025b) (off.) | 57.14 | 66.25 | 72.05 | 41.50 | 49.26 | 59.03 |
| Lp-Reg (off.) | 58.08 | 64.23 | 71.41 | 40.86 | 46.30 | 52.39 |
| GRPO (Shao et al., 2024a) (on.) | 55.19 | 63.93 | 70.48 | 42.86 | 49.90 | 57.85 |
| GRPO + Entropy Loss (on.) | 47.44 | 57.53 | 66.34 | 34.86 | 41.62 | 48.09 |
| 80/20 (Wang et al., 2025b) (on.) | 56.97 | 63.66 | 71.66 | 42.28 | 49.76 | 57.39 |
| Lp-Reg (on.) | **62.67**$_{+5.38\%}$ | **71.04**$_{+6.27\%}$ | **79.85**$_{+6.59\%}$ | **48.53**$_{+9.33\%}$ | **56.03**$_{+7.81\%}$ | **64.95**$_{+8.85\%}$ |

Table 4: Per-benchmark peak pass@k results on the challenging AIME24 and AIME25 benchmarks. Similar to Table 3, **scores reported denote the peak pass@k rate for each metric separately, implying they may be derived from different checkpoints.** Best scores are **bolded** and second-best scores are underlined. The relative improvement of Lp-Reg is indicated as a subscript.

## B.6 GENERALIZATION ACROSS MODELS AND DOMAINS

To further validate the generalizability of Lp-Reg, we extend our evaluation across different model architectures and domains.

**Extension to Llama3 Architecture** To assess effectiveness across various model architectures, we conduct experiments on Llama3-OctoThinker-8B (Wang et al., 2025c), a mid-trained model derived from Llama3-8B-Base (Grattafiori et al., 2024). The vanilla Llama3 series is known to present significant challenges for RLVR (Gandhi et al., 2025; Liu et al., 2025). As presented in Table 5, our proposed Lp-Reg outperforms all other methods by a substantial margin. Specifically, in on-policy training, Lp-Reg(on.) achieves an absolute gain of $2.88\%$ absolute accuracy over the second-best on-policy method, GRPO. For off-policy training, Lp-Reg(off.) demonstrates an even larger advantage, surpassing the nearest competing off-policy method by at least $3.62\%$ absolute accuracy.

These results strongly align with the findings observed on Qwen models, further highlighting the robustness of Lp-Reg across different foundational architectures.

**Domain Generalization: Science and Code** We also conduct comparative experiments across the science and code domains. For code generation, we train models on the AReaL-boba-2-RL-Code (Fu et al., 2025a) dataset and evaluate performance on the LCB-v5 (Jain et al., 2024) and LCB-v6 (Jain et al., 2024) benchmarks. For the science domain, the Qwen3-8B-Base model is trained on the SCP-116k (Lu et al., 2025) dataset, which covers biology, chemistry, and physics problems, and evaluated on the PhD-level GPQA-diamond (Rein et al., 2024) benchmark.

As shown in Table 6, Lp-Reg achieves the best overall performance in both the code and science domains. For the code generation task, Lp-Reg achieves the highest average score within both the on-policy and off-policy categories, respectively. On the challenging science task, Lp-Reg also demonstrates superior performance on the GPQA-diamond dataset, with Lp-Reg(on.) and Lp-Reg(off.) surpassing their respective baselines (GRPO(on.) and KL-Cov(off.)). The consistency of these improvements across mathematics, science, and code domains demonstrates the effectiveness and broad applicability of Lp-Reg.

| Method | AIME24 | AIME25 | Math-500 | Minerva | Olympiad Bench | Avg. |
|---|---|---|---|---|---|---|
| **Llama3-OctoThinker-8B** | | | | | | |
| GRPO (Shao et al., 2024a) (off.) | 4.38 | 4.58 | 60.00 | 26.47 | 25.93 | 24.27 |
| GSPO (Zheng et al., 2025a) (off.) | 4.58 | 2.50 | 58.80 | 29.41 | 25.33 | 24.13 |
| Clip-Higher (Yu et al., 2025) (off.) | 11.88 | 3.75 | 61.80 | 23.16 | 26.96 | 25.51 |
| KL-Cov (Cui et al., 2025) (off.) | 7.71 | 4.58 | 55.00 | 23.16 | 22.96 | 22.68 |
| 80/20 (Wang et al., 2025b) (off.) | 10.00 | 7.50 | 59.00 | 18.75 | 27.56 | 24.56 |
| Lp-Reg (off.) (ours) | 9.58 | 8.33 | 68.80 | 27.21 | 31.70 | 29.13 |
| GRPO (Shao et al., 2024a) (on.) | 15.42 | 12.50 | 76.20 | 33.09 | 43.26 | 36.09 |
| 80/20 (Wang et al., 2025b) (on.) | 11.67 | 4.17 | 73.60 | 27.21 | 37.48 | 30.82 |
| Lp-Reg (on.) (ours) | **18.33** | **16.88** | **79.00** | **35.29** | **45.33** | **38.97** |

Table 5: Main results on five mathematical reasoning benchmarks on **Llama3-OctoThinker-8B**. On-policy (on.) and off-policy (off.) training methods are highlighted with distinct colors. Benchmark scores correspond to the same checkpoint that achieves the highest average test set accuracy across the whole training. Best scores are **bolded** while second-best scores are underlined.

| Methods | Code | | | Science |
|---|---|---|---|---|
| | LCB-v5 | LCB-v6 | Avg. | GPQA-diamond |
| **Qwen3-8B-Base** | | | | |
| GRPO (Shao et al., 2024a)(off.) | 27.32 | 27.43 | 27.38 | 39.71 |
| GSPO (Zheng et al., 2025a)(off.) | 28.29 | 26.57 | 27.43 | 47.16 |
| Clip-Higher (Yu et al., 2025)(off.) | 27.10 | 27.57 | 27.34 | 48.61 |
| KL-Cov (Cui et al., 2025)(off.) | 28.74 | 27.43 | 28.09 | 49.18 |
| 80/20 (Wang et al., 2025b)(off.) | 26.57 | 27.64 | 27.11 | 45.90 |
| Lp-Reg(off.) | **29.57** | 27.57 | 28.57 | 51.77 |
| GRPO (Shao et al., 2024a)(on.) | 27.47 | 27.86 | 27.67 | 50.63 |
| 80/20 (Wang et al., 2025b)(on.) | 28.29 | 27.36 | 27.83 | 48.42 |
| Lp-Reg(on.) | 28.89 | **29.00** | **28.95** | **52.97** |

Table 6: Results on science and code domains on Qwen3-8B-Base. On-policy (on.) and off-policy (off.) training methods are highlighted with distinct colors. Benchmark scores correspond to the same checkpoint that achieves the highest average test set accuracy across the whole training. Best scores are **bolded** while second-best scores are underlined.

## B.7 COMPUTATIONAL OVERHEAD ANALYSIS

To analyze the computational overhead of Lp-Reg, particularly with large vocabularies, we analyze the complexity of its two core components: proxy distribution construction in Equation 5 and loss computation in Equation 6. We provide the PyTorch-style implementation for proxy distribution renormalization in Listing 1 and the Lp-Reg loss calculation in Listing 2.

```python
def forward_micro_batch(logits, kappa):
    # Standard Log-Softmax calculation
    log_prob = log_softmax(logits)

+   # 1. Calculate dynamic threshold
+   prob = exp(log_prob)
+   threshold = kappa * max(prob, axis=-1)

+   # 2. Filter noise
+   mask = prob < threshold
+   proxy_logits = logits.clone()
+   proxy_logits[mask] = -infinity

+   # 3. Re-normalization
+   proxy_log_prob = log_softmax(proxy_logits)

    return log_prob, proxy_log_prob
```

Listing 1: Pseudo-code of Proxy Distribution Construction

```python
def compute_policy_loss_lp_reg(old_log_prob, log_prob,
↪   proxy_log_prob, advantage, **args):
    # Standard PPO/GRPO Loss
    ratio = exp(log_prob - old_log_prob)
    pg_loss = maximum(-ratio * advantage, -clip(ratio) *
    ↪   advantage)

+   # 1. Identify tokens receiving negative feedback
+   neg_idx = indices(advantage < 0)

+   # 2. Select bottom rho% lowest probability tokens
+   k = int(len(neg_idx) * args["rho"])
+   low_prob_idx = topk(log_prob[neg_idx], k=k, largest=False)

+   # 3. Apply Regularization
+   mask = log_prob[low_prob_idx] < proxy_log_prob[low_prob_idx]
+   reg_idx = low_prob_idx[mask]

+   # 4. Calculate KL penalty term
+   pg_loss[reg_idx] += args["ppo_kl_coef"] * kl_penalty(
+       log_prob[reg_idx], proxy_log_prob[reg_idx])

    return pg_loss.mean()
```

Listing 2: Pseudo-code of Lp-Reg Loss Calculation

**Proxy Distribution Renormalization:** As shown in Listing 1, the renormalization process involves computing the maximum probability and re-evaluating log-probabilities. While these operations scale linearly with the vocabulary size $\mathcal{O}(|V|)$, they are structurally identical to the standard Softmax and Log-Softmax operations already required by the base model. These element-wise vector operations are highly parallelizable on GPUs and are memory-bandwidth bound rather than compute-bound. Consequently, their cost is negligible compared to the $\mathcal{O}(d_{model}^2)$ complexity of the Transformer's matrix multiplications, regardless of the vocabulary size.

**Loss Computation:** The regularization term requires identifying the lowest-probability tokens, which involves a Top-K selection (Listing 2). The computational complexity is $\mathcal{O}(N \log K)$ (using a heap) or $\mathcal{O}(N)$ (using QuickSelect), where $N$ is the total number of tokens in a micro-batch (typically $\approx 30,000$) and $K = \rho \cdot N$ ($\rho \approx 0.01$) is the number of selected tokens. Given that $N$ is relatively small and the operation is performed only once per optimization step (not during every inference step), this sorting overhead is computationally trivial.

**Empirical Verification:** We empirically validate this analysis by comparing the training runtime of GRPO and Lp-Reg in Table 7. To ensure a strictly fair comparison, we loaded checkpoints at 256, 512, and 768 steps and executed exactly one training update for each method under identical conditions of the same rollout data. The results show that Lp-Reg introduces a marginal overhead of approximately $0.3\% \sim 0.5\%$. This confirms that Lp-Reg is computationally lightweight and does not affect the scalability of training.

| Steps | Avg. Response Length | GRPO (s) | Lp-Reg (s) | Overhead |
|-------|---------------------|----------|-----------|----------|
| 256 | 4058.53 | 698.49 | 700.60 | +0.30% |
| 512 | 5794.25 | 973.74 | 978.62 | +0.50% |
| 768 | 6640.69 | 1137.24 | 1141.49 | +0.37% |

Table 7: Runtime comparison between GRPO and Lp-Reg under different training steps. Lp-Reg introduces only marginal overhead compared with GRPO.

## C  FURTHER ANALYSIS

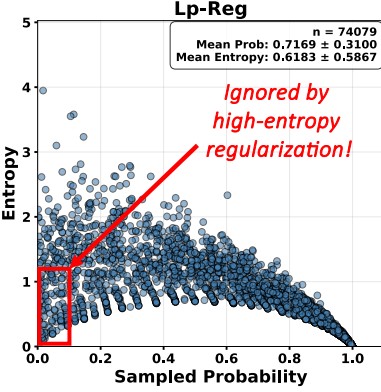

Figure 14: Probability–Entropy scatter plots of explorative tokens: "but", "wait", "perhaps", "alternatively", and "however". It displays a random sample of $5\%$ of all data points

### C.1  THEORETICAL DISCUSSION ON LOW-PROBABILITY VS. HIGH-ENTROPY TOKENS

While previous works have primarily utilized policy entropy as a proxy for exploration (Wang et al., 2025b), our approach distinguishes between high-entropy tokens and low-probability tokens. Empirical results presented in Table 2 and Figure 10 demonstrate that regularizing low-probability tokens yields significantly better stability and performance than targeting high-entropy tokens.

In this section, we provide a theoretical foundation for these results. We formally demonstrate that the set of tokens targeted by high-entropy methods is a *subset* of those captured by low-probability methods. Crucially, high-entropy strategies inherently overlook the region of low-probability tokens within low-entropy distributions, which is important for exploration, proven by empirical experiments.

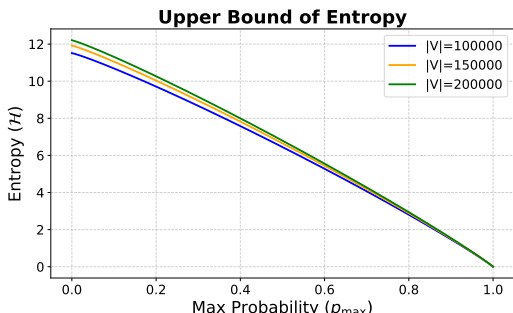

Figure 15: Theoretical bound of entropy $\mathcal{H}$ vs. max probability $p_{max} = \max_{o \in V} \pi_\theta(o|\cdot)$. The curve represents the maximum possible entropy for a given peak probability $p_{\max}$ with $|\mathcal{V}| = 100000$, $150000$, $200000$.

**Proposition 1** *Given the policy $\pi_\theta(\cdot|s)$ over a vocabulary $V$, and the policy entropy defined as $\mathcal{H}(\pi_\theta) = -\sum_{o \in V} \pi_\theta(o|s) \log(\pi_\theta(o|s))$, the following holds:*

$$\forall \epsilon \in (1/|V|, 1), \quad \exists \delta > 0, \quad s.t. \text{ if } \mathcal{H}(\pi_\theta) > \delta, \text{ then } \pi_\theta(o|s) < \epsilon, \quad \forall o \in V. \tag{7}$$

**Proof** Let $p_{\max} = \max_{o \in V} \pi_\theta(o|s)$ be the max token probability in the policy, and let $o^* = \arg\max_{o \in V} \pi_\theta(o|s)$. Accordingly, $\pi_\theta(o^*|s) = p_{\max}$.

Firstly, we decompose the entropy term by separating the maximal probability token $o^*$ from the rest of the vocabulary $V \setminus \{o^*\}$:

$$\mathcal{H}(\pi_\theta) = -p_{\max} \log p_{\max} - \sum_{o \neq o^*} \pi_\theta(o|s) \log \pi_\theta(o|s). \tag{8}$$

Let $K = |V| - 1$. The remaining probability mass is $1 - p_{\max}$. Since $f(x) = x \log x$ is a convex function, according to Jensen's Inequality, the entropy of the remaining tokens is maximized when the distribution is uniform, i.e., $\pi_\theta(o|s) = \frac{1-p_{\max}}{K}$ for all $o \neq o^*$. Substituting this into the equation, we obtain the upper bound function $g(p_{\max})$:

$$\mathcal{H}(\pi_\theta) \leq -p_{\max} \log p_{\max} - \sum_{o \neq o^*} \frac{1 - p_{\max}}{K} \log \left( \frac{1 - p_{\max}}{K} \right)$$

$$= -p_{\max} \log p_{\max} - (1 - p_{\max}) \log \left( \frac{1 - p_{\max}}{K} \right) \triangleq g(p_{\max}). \tag{9}$$

Then, we analyze the monotonicity of the function $g(x) = -x \log x - (1 - x) \log \frac{1-x}{K}$ for $x \in (1/|V|, 1)$. Taking the derivative with respect to $x$:

$$g'(x) = -(\log x + 1) - \left[ (-1) \cdot \log \left( \frac{1 - x}{K} \right) + (1 - x) \cdot \frac{K}{1 - x} \cdot \left( -\frac{1}{K} \right) \right]$$

$$= -\log x - 1 + \log \left( \frac{1 - x}{K} \right) + 1$$

$$= \log \left( \frac{1 - x}{Kx} \right). \tag{10}$$

Since $K = |V| - 1$, we have $\frac{1-x}{Kx} < 1$ for any $x > \frac{1}{|V|}$. Thus, $g'(x) < 0$ when $x \in (\frac{1}{|V|}, 1)$, which means $g(x)$ is strictly monotonically decreasing on the interval $(\frac{1}{|V|}, 1)$.

Finally, Let $\delta = g(\epsilon)$. Since $\epsilon \in (1/|V|, 1)$, $\delta$ is a well-defined positive value. Assume the condition $\mathcal{H}(\pi_\theta) > \delta$ holds. By the upper bound established above, we have:

$$g(p_{\max}) \geq \mathcal{H}(\pi_\theta) > \delta = g(\epsilon). \tag{11}$$

Thus, $g(p_{\max}) > g(\epsilon)$. Since we have proved that $g(x)$ is strictly monotonically decreasing for $x > 1/|V|$, the inequality of function values implies the reverse inequality of arguments:

$$p_{\max} < \epsilon. \tag{12}$$

By definition, $\pi_\theta(o|s) \leq p_{\max}$ for all $o \in V$. Therefore, $\pi_\theta(o|s) < \epsilon$ for all $o \in V$.

Proposition 1 theoretically establishes that high entropy strictly implies low probability for all tokens. In other words, the set of tokens targeted by high-entropy methods is almost a subset of those targeted by low-probability regularization.

However, the converse does not hold. Low-probability tokens can be sampled not only from high-entropy positions but also from low-entropy positions. The latter scenario constitutes a blind spot for entropy-based methods: when the model is in a low entropy position, entropy methods ignore the step. Yet, as shown in Figure 14, valuable explorative tokens (e.g., "but", "wait") frequently appear in this low-probability, low-entropy regime. The theoretical upper bound visualized in Figure 15 further confirms that entropy maximization is restricted to the left-most region, whereas our Lp-Reg remains effective across the entire region. This explains why Lp-Reg outperforms high-entropy regularization, as validated by our experiments.

## C.2 TRAJECTORY-LEVEL TOKEN ANALYSIS

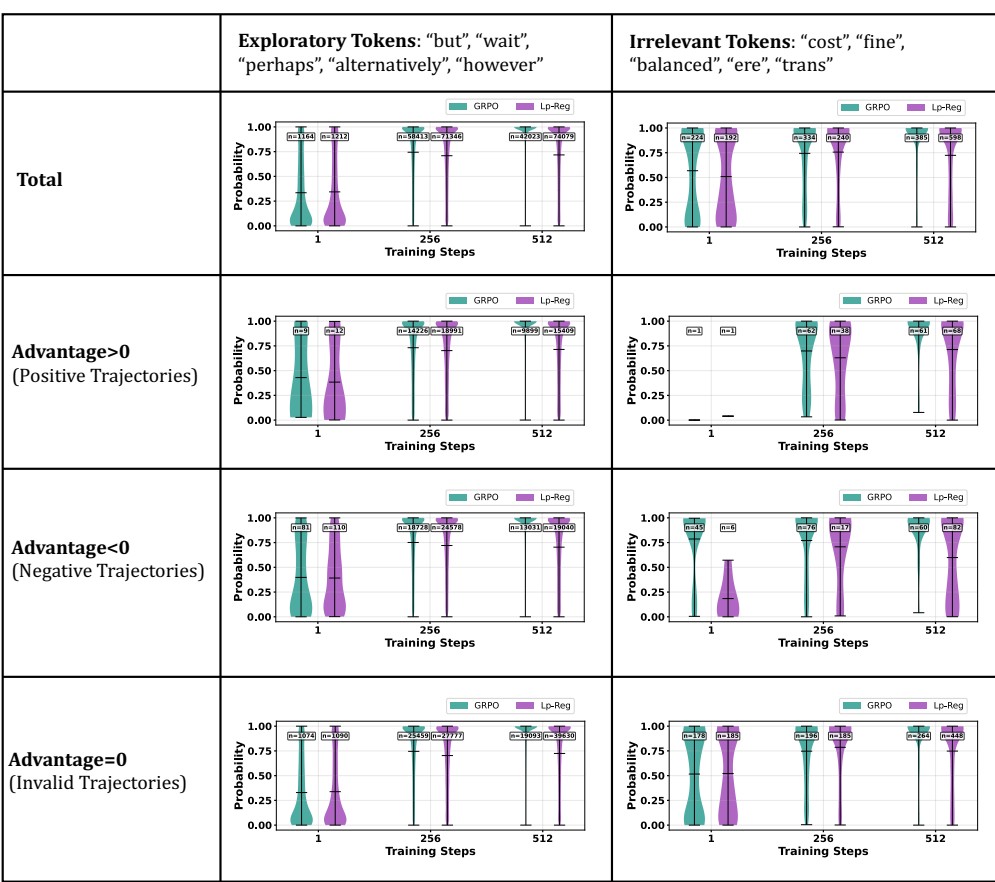

Figure 16: Trajectory-level probability analysis distinguishing exploratory tokens (left) from irrelevant tokens (right). The distributions are decomposed into positive ($A > 0$), negative ($A < 0$), and invalid ($A = 0$) trajectories, where $n$ represents the sampling token number.

In this section, we conduct a fine-grained trajectory-level analysis to characterize the sampling probability distributions of specific tokens. We decompose the token sampling distributions based on the advantage values of their corresponding trajectories: positive ($A_i > 0$), negative ($A_i < 0$), and neutral/invalid ($A_i = 0$). The comparative results between exploratory tokens (e.g., "but", "wait") and irrelevant tokens (e.g., "cost") are visualized in Figure 16.

As shown in Figure 16, we observe that the probability distributions of exploratory tokens are re-markably similar across Positive and Negative trajectories, under both standard GRPO and Lp-Reg. This indicates that these tokens function as reasoning patterns: they represent the mechanism of the reasoning attempt, rather than the determinant of the final outcome. Just as scratchpad paper is utilized for both correct and incorrect solutions, a negative trajectory containing "wait" repre-sents a failed reasoning attempt, which is fundamentally different from a failure due to a lack of reasoning. This is further corroborated by the contrast in sampling density between active learning groups ($A \neq 0$) and static groups ($A = 0$). The former exhibits a significantly higher density of low-probability tokens, while the latter shows much less. This is consistent with the intuition that ac-tive exploration yields diverse outcomes (both successes and failures), whereas a lack of exploration leads to concentrated, often stagnant results. Because these tokens appear abundantly in negative tra-jectories simply due to the high volume of failed attempts during exploration, standard GRPO tends to systematically suppress them. Lp-Reg successfully preserves these essential patterns, ensuring the model retains the capacity to reason even when individual attempts fail.

Importantly, a distinct divergence emerges when comparing standard GRPO and Lp-Reg. As illus-trated in Figure 16 (Step 512), standard GRPO exhibits a significant reduction in low-probability token sampling in later training stages, signaling a diminishing of exploration attempts when uncer-tain (low probability). This collapse directly corresponds to the performance bottleneck observed in Figure 17. In contrast, Lp-Reg maintains robust low-probability token sampling throughout long-horizon training, coinciding with a continuously increasing accuracy score. This demonstrates the effectiveness of Lp-Reg in sustaining exploration.

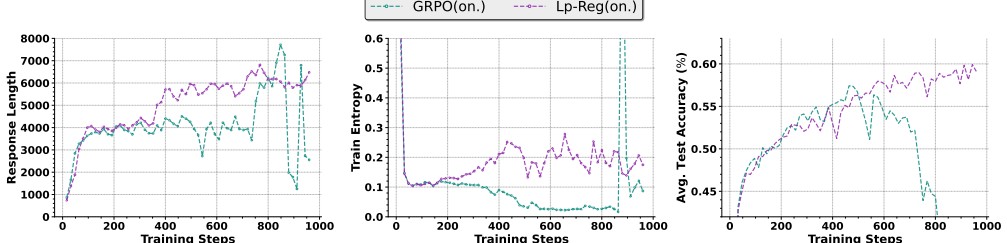

Figure 17: Comparison between standard GRPO and Lp-Reg on Qwen3-14B-Base.

## C.3 DISCUSSION ON LOW-PROBABILITY TOKENS

In this section, we discuss the difference between our Lp-Reg and Lopti (Yang et al., 2025b), a recent work that also investigates low-probability tokens. It is important to note that Lp-Reg and Lopti are not in conflict; rather, they identify and address two distinct orthogonal challenges in RLVR training. Lopti focuses on improving gradient dynamics for better data efficiency, while Lp-Reg focuses on ensuring long-horizon training stability. The distinction is from three perspectives: the core research problem, the methodological approach, and new, direct experimental comparisons.

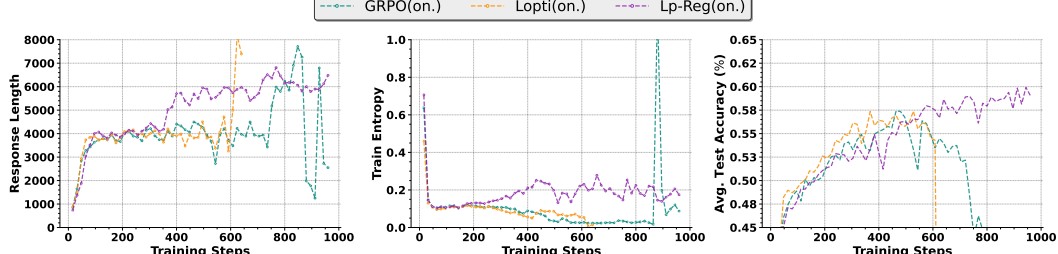

Figure 18: Comparison on standard GRPO, Lopti, and Lp-Reg.

(1) Different Core Research Problems: Lopti focuses on the training efficiency, whereas Lp-Reg focuses on the training stability. These represent two orthogonal axes of improvement for RLVR. In detail, Lopti identifies that low-probability tokens generate gradients with disproportionately large norms. Its core focus is on how this "over-domination" suppresses gradient signals from high-probability tokens, thereby reducing the data efficiency of the training process. In contrast, our Lp-Reg identifies the systematic elimination of low-probability tokens with exploratory semantics (e.g., "wait"), which we term "*reasoning sparks*". Our core focus is on how the over-penalization of these tokens leads to a loss of exploration capacity with the entropy collapse phenomenon, hindering the model from achieving higher performance in long-horizon stable training.

(2) Different Methodological Approaches: Lopti's method of separate gradient updates and Lp-Reg's regularization are distinct and non-conflicting algorithms. Specifically, to prevent large-norm gradients from low-probability tokens suppressing signals from high-probability tokens, Lopti separates the loss computation for these two groups and updates the model parameters twice per micro-batch. For another goal to protect low-probability tokens from over-penalization in RLVR, Lp-Reg introduces a regularization on them via a KL divergence between the current policy and a filtered proxy policy.

(3) Empirical evidence from long-horizon experiments: To empirically validate our claims, we have conducted a long-horizon training experiment comparing Lopti, Lp-Reg, and the GRPO baseline for 1,000 steps. As shown in Figure 18, Lopti shows a faster initial rise in test accuracy, confirming its effectiveness at accelerating learning, consistent with the findings in their paper. However, after approximately 600 steps, Lopti's performance plateaus, and its training entropy collapses in the same manner as the GRPO baseline. This shows that improving data efficiency does not inherently solve the long-term exploration problem. In contrast, our Lp-Reg demonstrates stable performance improvement throughout the 1,000 steps, correlated with its ability to maintain policy entropy. This sustained exploration allows it to achieve a significantly higher final accuracy.

In conclusion, Lp-Reg and the Lopti study address distinct, orthogonal challenges in RLVR. The choice between these methods may depend on the specific training objectives. While investigating a potential combination could be an interesting avenue for future research, our primary contribution here is to formally identify the exploration stability and provide an effective solution for it. We have added this detailed comparison to our revised manuscript to contextualize our work better and highlight its unique conceptual contribution.

## C.4 DETAILS OF SAMPLING PROBABILITY DENSITY

This section provides a detailed, token-by-token breakdown of the aggregated distributions presented in Figure 1c and Figure 1d of the main paper, reinforcing the conclusions drawn from our analysis.

Figure 20 exhibits the individual distribution of observed sampling probabilities for meaningful exploratory tokens, also known as *reasoning sparks*: "but", "wait", "perhaps", "alternatively", and "however". These tokens are also frequently analyzed as representative cases in previous studies (DeepSeek-AI et al., 2025; Muennighoff et al., 2025; Hu et al., 2025; Qian et al., 2025; Wang et al., 2025b). A consistent trend is observable across all five tokens, validating our claims in the introduction. With standard GRPO training, the ability to sample these tokens at low probabilities is systematically eliminated, causing their distributions to collapse and shift towards higher probabilities. The indiscriminate entropy bonus (GRPO + Entropy Loss) is largely ineffective at restoring this crucial low-probability tail. In stark contrast, our proposed method, Lp-Reg, consistently maintains a healthy, wide distribution for each of these tokens, demonstrating its effectiveness in preserving the model's capacity for nuanced exploration.

Conversely, Figure 21 details the behavior of a class of meaningless noise tokens: "cost", "fine", "balanced", "ere", and "trans". These individual plots clearly illustrate the detrimental side effect of a simple entropy bonus. For nearly every token, the GRPO + Entropy Loss baseline significantly amplifies the sampling of this irrelevant noise, which, as shown in our main analysis, contributes to a faster performance collapse. Lp-Reg, by design, avoids this amplification and maintains a suppressed probability distribution for these tokens, comparable to or even more constrained than the standard GRPO baseline.

These detailed visualizations confirm that the phenomena of reasoning spark elimination and noise amplification are not artifacts of aggregation but are consistent patterns at the individual token level. This provides strong, granular evidence for the central challenge our paper addresses and highlights the necessity of a selective preservation mechanism like Lp-Reg.

### C.5  DETAILS OF PROBABILITY-ENTROPY DISTRIBUTION

To supplement the aggregated analysis presented in Figure 6 of the main text, this section provides a detailed breakdown of the probability-entropy distributions for individual *reasoning sparks*. Figure 22 shows a consistent pattern across all representative tokens, ranging from "but" (Figure 22a) to "however" (Figure 22e). For frequently occurring tokens such as "but", "wait", and "perhaps", we randomly subsample one out of every 20 instances for visualization. Under the baseline GRPO, these sparks are consistently confined to a low-entropy, high-probability region, indicating a collapse into deterministic usage. In contrast, the addition of an entropy loss pushes these tokens into highly scattered, often excessively high-entropy states, suggesting an uncontrolled and potentially noisy form of exploration. Our method, Lp-Reg, strikes a crucial balance, maintaining a structured and broad distribution across a healthy range of entropy values. This consistent behavior demonstrates that the trends identified in the aggregated data are not artifacts of averaging. The individual plots offer strong, disaggregated evidence for our central claim: Lp-Reg effectively preserves the exploratory potential of reasoning sparks by preventing both the deterministic collapse seen in the baseline and the chaotic scattering induced by the indiscriminate entropy bonus.

### C.6  TRAINING DYNAMICS OF REGULARIZED TOKEN

To better understand how Lp-Reg operates during training, we analyze the dynamics of the probability threshold $\delta_\rho^{\mathcal{B}}$ and the proportion of low-probability tokens subjected to regularization. As shown in Figure 19, the threshold $\delta_\rho^{\mathcal{B}}$ gradually decreases with training steps. In parallel, the regularization ratio also declines steadily. This trend suggests that as training progresses, an increasing share of extremely low-probability tokens correspond to meaningless noise, while the semantically meaningful tokens are lifted into higher-probability regions and thus require less regularization.

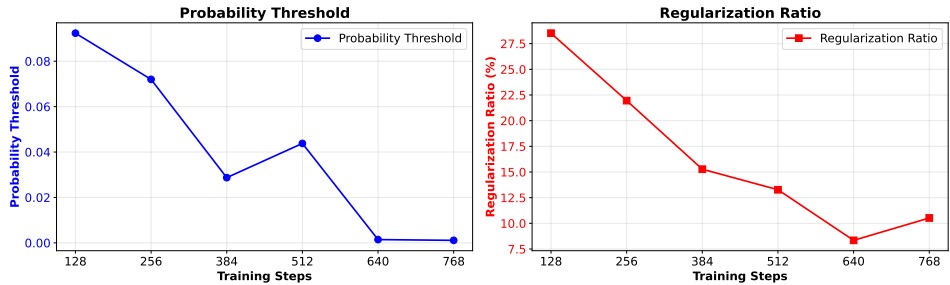

Figure 19: Training dynamics of the probability threshold and regularization ratio.

### C.7  CASE STUDY

To further illustrate the effect of the filter applied on low-probability tokens, Figure 23 to Figure 25 presents a case study of a model-generated response, where low-probability tokens are highlighted according to whether they were preserved or filtered. Tokens with probability greater than $\tau$ are those retained by the filter, while tokens with probability smaller than $\tau$ are suppressed. The preserved tokens include meaningful exploratory markers such as "Then", "Wait", which guide the reasoning trajectory, whereas the discarded set largely consists of noisy tokens such as "We", "also", "that". This qualitative evidence complements our quantitative analysis, demonstrating that Lp-Reg effectively leverages min-p distribution re-normalization to reliably distinguish between semantically meaningful exploratory reasoning sparks and destabilizing noise.

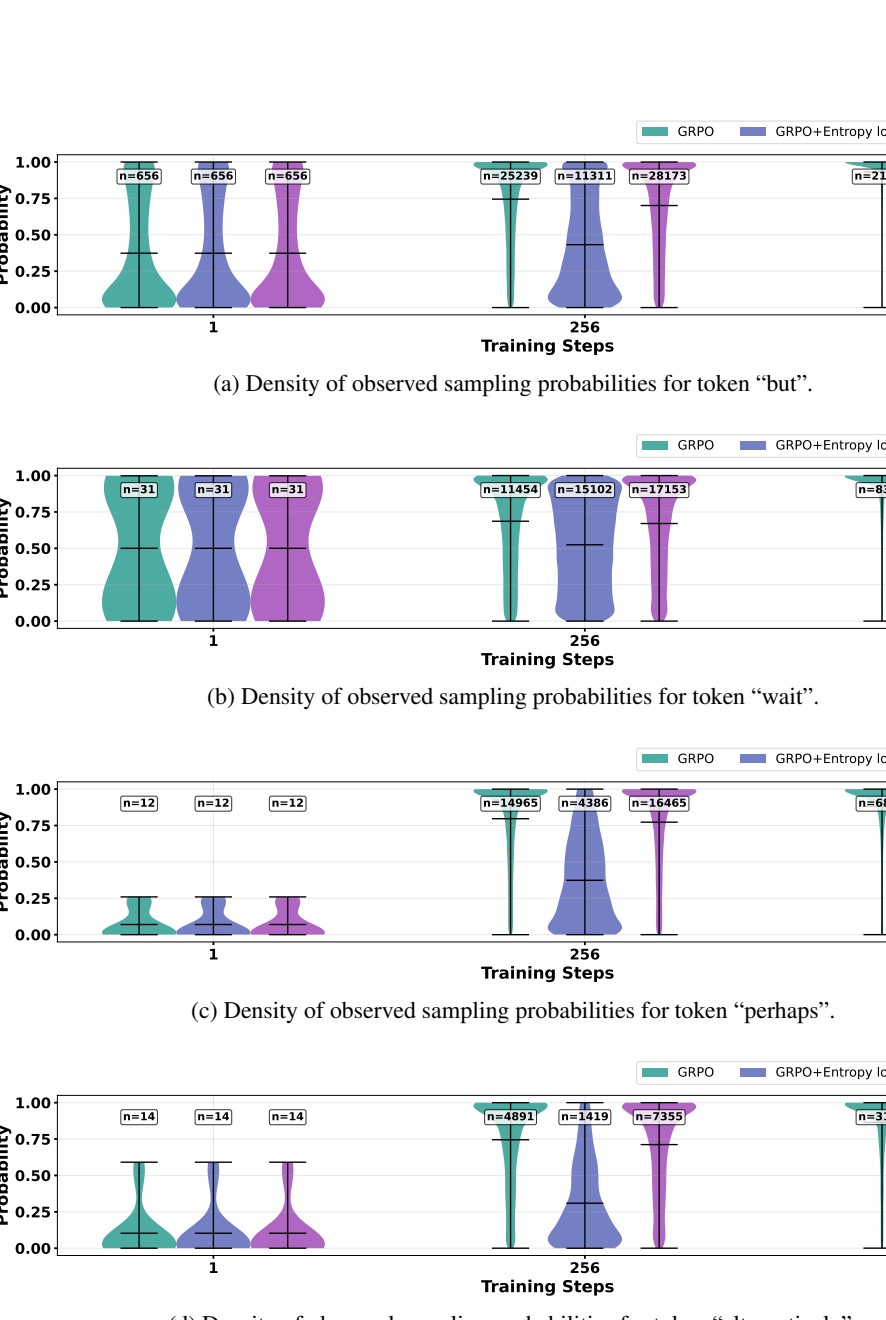

(a) Density of observed sampling probabilities for token "but".

(b) Density of observed sampling probabilities for token "wait".

(c) Density of observed sampling probabilities for token "perhaps".

(d) Density of observed sampling probabilities for token "alternatively".

(e) Density of observed sampling probabilities for token "however".

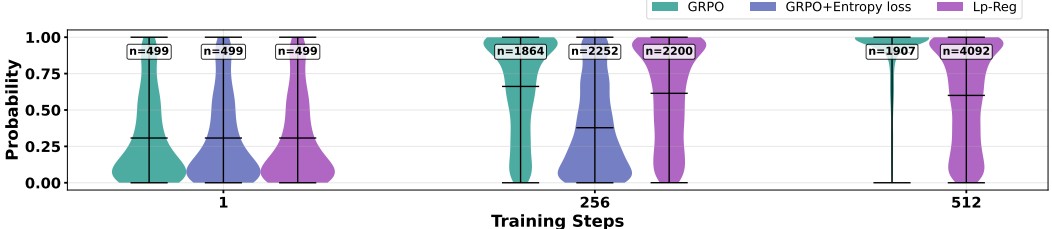

Figure 20: Individual Density of observed sampling probabilities for meaningful exploratory tokens: "but", "wait", "perhaps", "alternatively", and "however".

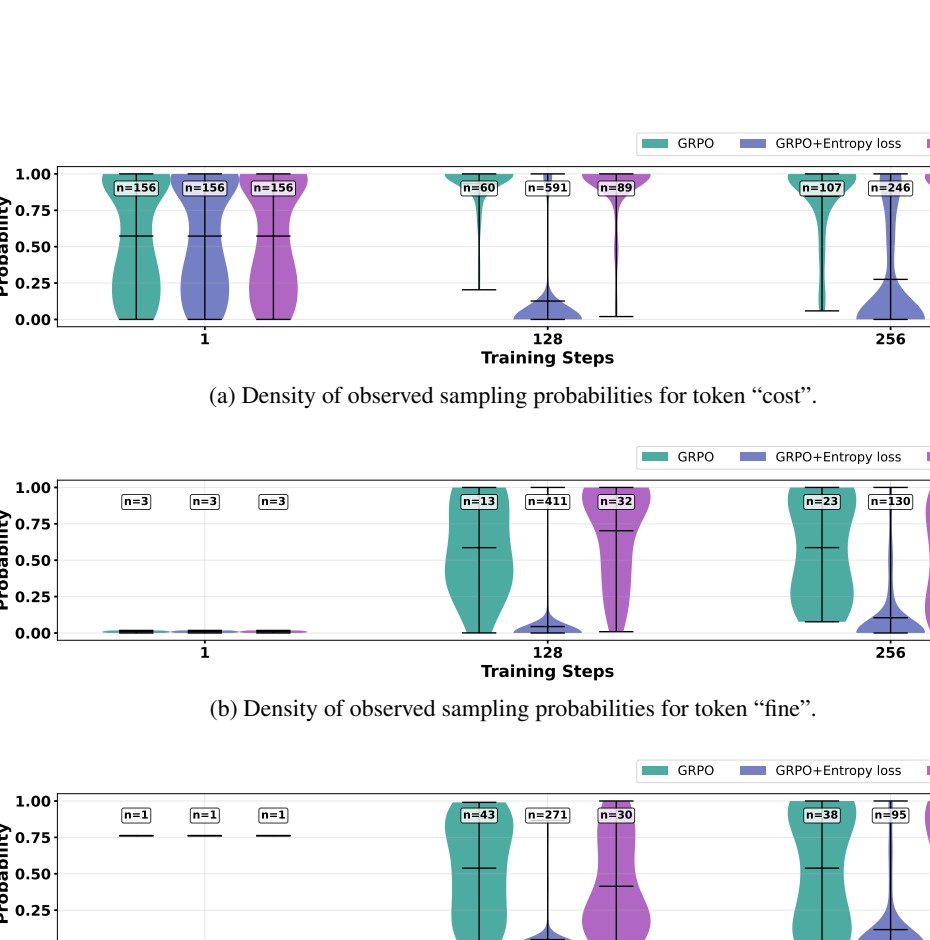

(a) Density of observed sampling probabilities for token "cost".

(b) Density of observed sampling probabilities for token "fine".

(c) Density of observed sampling probabilities for token "balanced".

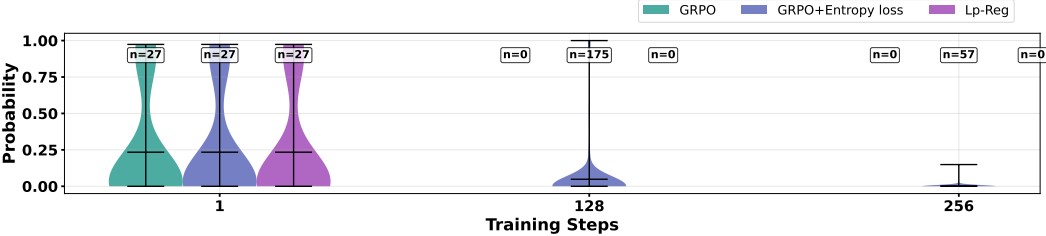

(d) Density of observed sampling probabilities for token "ere".

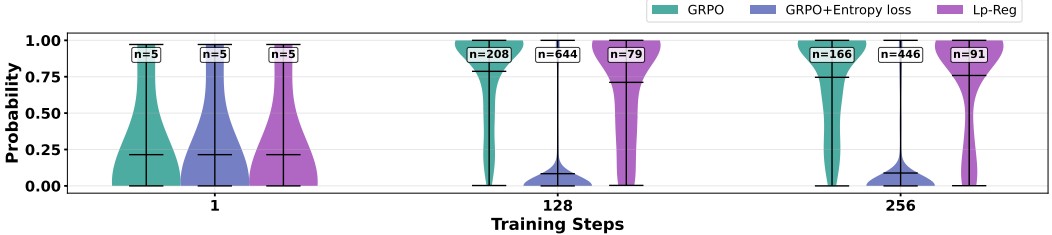

(e) Density of observed sampling probabilities for token "trans".

Figure 21: Individual Density of observed sampling probabilities for meaningless tokens: "cost", "fine", "balanced", "ere", and "trans".

(a) Scattered probability–entropy plot of observed sampling instances for the token "but".

(b) Scattered probability–entropy plot of observed sampling instances for the token "wait".

(c) Scattered probability–entropy plot of observed sampling instances for the token "perhaps".

(d) Scattered probability–entropy plot of observed sampling instances for the token "alternatively".

(e) Scattered probability–entropy plot of observed sampling instances for the token "however".

Figure 22: Individual scattered probability–entropy plot of observed sampling instances for meaningful tokens: "but", "wait", "perhaps", "alternatively", and "however".

**Single Response Case Study**
**Lowest Probability Tokens threshold:** $\delta_\rho^B = 0.026208$
**(Part 1 of 3)**

■ Reserved low-probability tokens: $\pi_\theta < \delta_\rho^B$ and $\pi_\theta > \tau$
■ Filtered low-probability tokens: $\pi_\theta < \delta_\rho^B$ and $\pi_\theta < \tau$
■ Normal tokens: $\pi_\theta > \delta_\rho^B$

Assistant :\n Alright , I have this problem about a square that 's divided into smaller squares , and the process keeps repeating indefinitely . Some parts are shaded , and I need to find out what fraction of the whole figure is shaded . The final answer should be in the form of a fraction \( \frac { m }{ n }\ ), and I need to provide the value of \( m + n \ ). Let 's break this down step by step .\n\n ### Understanding the Problem \n\n First , let 's try to understand what 's happening :\n\n 1 . ** Initial Square **: We start with one big square .\n 2 . ** First Division **: This square is divided into nine smaller squares of equal area . So , it 's like a 3 x 3 grid .\n 3 . ** Center Division **: The center square of this 3 x 3 grid is then further divided into nine even smaller squares ( 3 x 3 again ).\n 4 . ** Pattern Contin ues **: This process repeats indefinitely âĢĶ each new center square is divided into nine smaller squares .\n\n From the As ym pt ote code , it seems that at each step , certain parts **(** corn ers ) of the squares are shaded . Specifically , the four corner squares of each smaller subdivision are shaded .\n\n Wait , looking at the code :\n\n ``` asy
fill ((- n ,n )-- (- 1 / 3 *n ,n )-- (- 1 / 3 *n , 1 / 3 *n )-- (- n , 1 / 3 *n )-- cycle );\n fill ((- n ,- n )-- (- 1 / 3 *n ,- n )-- (- 1 / 3 *n ,- 1 / 3 *n )-- (- n ,- 1 / 3 *n )-- cycle );\n fill (( n ,- n )-- ( 1 / 3 *n ,- n )-- ( 1 / 3 *n ,- 1 / 3 *n )-- ( n ,- 1 / 3 *n )-- cycle );\n fill (( n ,n )-- ( 1 / 3 *n ,n )-- ( 1 / 3 *n , 1 / 3 *n )-- ( n , 1 / 3 *n )-- cycle );\n ``` \n\n This code seems to shade the four corner squares of each subdivision at each level . So , at each step , four **of** the nine smaller squares are shaded .\n\n ### Visual Representation \n\n Let 's try to visualize this :\n\n 1 . ** Level 0 **: The original big square .\n 2 . ** Level 1 **: Div ided into 9 squares , and the four corners are shaded .\n 3 . ** Level 2 **: The center square of Level 1 is divided into 9 even smaller squares , and its four corners are shaded .\n 4 . ** Level 3 **: The center of **that latest** division is divided again , and so on , infinitely .\n\n Wait , actually , looking back at the As ym pt ote code , it seems that at every level , the four corner squares of each **current** smallest subdivision are shaded . But in the description , it mentions the center square is divided , not necessarily the corners . Maybe I mis interpreted .\n\n Wait , the code **sh** ad ings correspond to the corners of each subdivision , but the description says the center square is divided . Hmm , perhaps the shaded parts are always the four corners of the current center square .\n\n But perhaps it 's better to assign a value .\n\n ### Assign ing Areas \n\n Let 's assign an area to the original square to make calculations easier . Let 's say the area of the original square is 1 .\n\n - ** Level 0 **: Total area = 1 . Fraction shaded = 0 ? Wait , no , according to the As ym pt ote code , the parts shaded are the four corners of each subdivision , not clearly defined yet .\n\n Wait , perhaps it 's better to think in terms of geometric series .\n\n At each level :\n\n - The entire figure is divided into 9 equal smaller squares .\n - Out of these , 4 are shaded ( the corners ).\n\n Wait , perhaps it 's not exactly that . Looking at the code , for each square at a certain level , its four corners in the next subdivision are shaded . But the center square continues to be subdiv ided .\n\n Wait , perhaps it 's better to think recursively .\n\n ### Recursive Thinking \n\n Let 's denote :\n\n - \( S _n \ ) as the total shaded area after \( n \ ) levels of subdivision .\n\n But since it 's infinite , we need to consider the limit as \( n \ ) approaches infinity .\n\n At each step :\n\n - The current **smallest** squares are divided into 9 even smaller squares .\n - Out of these 9 , 4 are shaded .\n\n Wait , perhaps it 's better to think of the entire figure and how much is shaded at each stage .\n\n Let 's consider the first level :\n\n - Original square is divided into 9 .\n - Sh aded area : the four corners , each **has** area \( \ frac { 1 }{ 9 } \ ), so total shaded at level 1 : \( 4 \ times \ frac { 1 }{ 9 } = \ frac { 4 }{ 9 } \ ).\n\n Wait , but then the center square is further divided , and in its **corners** , etc .\n\n Wait , perhaps it 's better to think of it as :\n\n At each subdivision , for each existing square , we shade four of its nine smaller parts , and then the center is subdiv ided further .\n\n This seems like it might lead to a geometric series where at each step , a fraction **is** added to the shaded area .\n\n But I 'm not entirely clear on the pattern . Maybe I should try to compute the shaded area at each step .\n\n ### Computing Sh aded Area \n\n Let 's try to compute the shaded area after a few levels to see if a pattern emerges .\n\n ** Level 0 **: **Sh** aded area = 0 .\n\n ** Level 1 **: The original square is divided into 9 , and 4 are shaded ⸤

Figure 23: An Example generated by Qwen3-14B-Base model trained by Lp-Reg from math reasoning. (Part 1)

**Single Response Case Study**
**Lowest Probability Tokens threshold:** $\delta_\rho^B = 0.026208$
**(Part 2 of 3)**

Reserved low-probability tokens: $\pi_\theta < \delta_\rho^B$ and $\pi_\theta > \tau$
Filtered low-probability tokens: $\pi_\theta < \delta_\rho^B$ and $\pi_\theta < \tau$
Normal tokens: $\pi_\theta > \delta_\rho^B$

\( S _ 1 = \ frac { 4 }{ 9 } \ \).\n\n Wait , but then the process continues . The center square ( of area \( \ frac { 1 }{ 9 } \ )) is further divided into 9 squares , each of area \( \ frac { 1 }{ 8 1 } \ \ ), and 4 of these are shaded : \( \ frac { 4 }{ 8 1 } \ \).\n\n So , total shaded after Level 2 : \( S _ 2 = \ frac { 4 }{ 9 } + \ frac { 4 }{ 8 1 } \ \).\n\n Similarly , the center of this latest subdivision ( which was a square of area \( \ frac { 1 }{ 8 1 } \ )) has a center that gets divided further , and so on .\n\n Wait , perhaps it 's better to think of it as an infinite series .\n\n Each time we divide a square into 9 and shade 4 , and then the center is divided again .\n\n This seems like a geometric series where each term is a fraction of the previous shaded area .\n\n Wait , let 's see :\n\n At Level 1 : shaded area \( \ frac { 4 }{ 9 } \ \).\n\n At Level 2 : shaded area \( \ frac { 4 }{ 8 1 } \ \ ) ( from the center square at Level 1 ).\n\n At Level 3 : shaded area \( \ frac { 4 }{ 7 2 9 } \ \ ) ( from the center square at Level 2 ), and so on .\n\n Wait , perhaps the total shaded area is the sum of those :\n\n \( S = \ frac { 4 }{ 9 } + \ frac { 4 }{ 8 1 } + \ frac { 4 }{ 7 2 9 } + \ cd ots \ )\n\n This is a geometric series with first term \( a = \ frac { 4 }{ 9 } \ ) and common ratio \( r = \ frac { 1 }{ 9 } \ \).\n\n The sum of an infinite geometric series is \( S = \ frac { a }{ 1 - r } \ \).\n\n So , \( S = \ frac {\ frac { 4 }{ 9 }} { 1 - \ frac { 1 }{ 9 }} = \ frac {\ frac { 4 }{ 9 }} {\ frac { 8 }{ 9 }} = \ frac { 4 }{ 8 } = \ frac { 1 }{ 2 } \ \).\n\n Wait , but I 'm not entirely confident this captures the entire shaded area because the shaded parts are at each subdivision level , and the pattern might be different .\n\n Wait , perhaps I should consider that at each subdivision , the shaded parts are not just in the outer squares but are nested within the subdivisions .\n\n Alternatively , perhaps the correct approach is to consider that at each step , the shaded area added is \( \ frac { 4 }{ 9 } \ ) of the area being subdiv ided , and the un sh aded center continues to be subdiv ided .\n\n Wait , perhaps a better approach is to consider the self -s im ilarity of the problem .\n\n ### Self -S im ilarity Approach \n\n Let 's denote \( S \ ) as the total shaded area of the original square .\n\n Looking at the first subdivision :\n\n - We have 9 smaller squares .\n - 4 of them are shaded immediately : each has area \( \ frac { 1 }{ 9 } \ ), so total \( \ frac { 4 }{ 9 } \ ).\n - The center square ( also of area \( \ frac { 1 }{ 9 } \ )) is a smaller version of the original figure , and its shaded area would be \( \ frac { S }{ 9 } \ ).\n\n Wait , no . Actually , the center square is undergoing the same shading of its corners initially ( in the context of the original shading ).\n\n Wait , perhaps it 's better to think that the entire figure maintains self -s im ilarity where :\n\n Total shaded area \( S = \ ) shaded in first level \( + \ ) shaded in scaled -down version \( = \ frac { 4 }{ 9 } + \ frac { S }{ 9 } \ \).\n\n Wait , let 's think carefully :\n\n At the first level :\n\n - Sh aded area : \( \ frac { 4 }{ 9 } \ ) ( the four corner squares ).\n - The center square ( of area \( \ frac { 1 }{ 9 } \ )) also follows the same pattern , but in its divisions , its corners will be shaded in the next level , and so on .\n\n Wait , perhaps the recursive relation is :\n\n \( S = \ frac { 4 }{ 9 } + \ frac { S }{ 9 } \ \).\n\n Wait , let 's see :\n\n The total shaded area is the shaded area from the first subdivision plus the shaded area from the pattern that would have been applied to the center square if it were the original , but scaled down by a factor of \( \ frac { 1 }{ 9 } \ ).\n\n Wait , but in the first subdivision , only the outer squares are immediately shaded , and the center continues to be subdiv ided indefinitely .\n\n Wait , perhaps it 's better to think of it as :\n\n At each step , we shade four out of nine parts of each square at the current level , and then the center square is treated as a new " original " square at the next level .\n\n Thus , the total shaded area \( S \ ) satisfies :\n\n \( S = \ frac { 4 }{ 9 } + \ frac { S }{ 9 } \ \).\n\n Wait , let 's verify :\n\n The first term \( \ frac { 4 }{ 9 } \ ) accounts for the four shaded corner squares at the first level .\n\n Then , the center square ( of area \( \ frac { 1 }{ 9 } \ )) has its own version of the problem , so its shaded area is \( \ frac { S }{ 9 } \ ).\n\n Thus , the equation \( S = \ frac { 4 }{ 9 } + \ frac { S }{ 9 } \ ) seems correct .\n\n Now , solve for \( S \ ):\n\n \( S - \ frac { S }{ 9 } = \ frac { 4 }{ 9 } \ )\n\n \( \ frac { 8 S }{ 9 } = \ frac { 4 }{ 9 } \ )\n\n Multiply both sides by 9 :\n\n \( 8 S = 4 \ )\n\n Div ide both sides by 8 :\n\n \( S = \ frac { 4 }{ 8 } = \ frac { 1 }{ 2 } \ ).\n\n So , the total shaded fraction is \( \ frac { 1 }{ 2 } \ ).\n\n But the problem states that the answer should be in the form \( \ frac { m }{ n } \ ) where \( \ gcd ( m , n ) = 1 \ ), and we need to

Figure 24: An Example generated by Qwen3-14B-Base model trained by Lp-Reg from math reasoning. (Part 2)

**Single Response Case Study**
**Lowest Probability Tokens threshold:** $\delta_\rho^B = 0.026208$
**(Part 3 of 3)**

Reserved low-probability tokens: $\pi_\theta < \delta_\rho^B$ and $\pi_\theta > \tau$
Filtered low-probability tokens: $\pi_\theta < \delta_\rho^B$ and $\pi_\theta < \tau$
Normal tokens: $\pi_\theta > \delta_\rho^B$

find \( m + n \ ).\n\n Here , \( \ frac { m }{ n } = \ frac { 1 }{ 2 } \ ), so \( m = 1 \ ), \( n = 2 \ ), and **thus** \( m + n = 3 \ ).\n\n Wait , but before final izing , let 's double -check the recursive relation .\n\n ### Ver ifying the Recursive Relation \n\n Let 's consider a simpler case to verify :\n\n Sup pose we only do the first level of subdivision . Then , the shaded area is \( \ frac { 4 }{ 9 } \ ).\n\n According to our recursive relation \( S = \ frac { 4 }{ 9 } + \ frac { S }{ 9 } \ ), if we only do **level** 1 , then perhaps the \( \ frac { S }{ 9 } \ ) should be zero , but our initial assumption **that** \( S = \ frac { 4 }{ 9 } + \ frac { S }{ 9 } \ ) holds for infinite subdivisions .\n\n Alternatively , perhaps it 's better to think that at each level , the shaded area added is **correctly** captured by \( \ frac { 4 }{ 9 } \ ) of the current area being considered , and the process continues similarly .\n\n Given that , our calculation seems consistent .\n\n ### Final Answer \n\n The fractional part of the figure that is shaded is \( \ frac { 1 }{ 2 } \ ). Therefore , \( m = 1 \ ), \( n = 2 \ ), and \( m + n = 3 \ ).\n\n **Wait** , but looking back at the problem , it mentions that the answer is in the form \( \ frac { m }{ n } \ ) with \( gcd (m , n ) = 1 \ ), and we need to find \( m + n \ ). Here , \( \ frac { 1 }{ 2 } \ ) satisfies this with \( m = 1 \ ), \( n = 2 \ ), so \( m + n = 3 \ ).\n\n But before final izing , I recall that sometimes such problems have the shaded area as \( \ frac { 5 }{ 12 } \ ), but based on our reasoning , it 's \( \ frac { 1 }{ 2 } \ ). **Double** -check ing the recursive relation seems correct .\n\n Alternatively , perhaps the shaded parts are not just the immediate corners but also within the subdivisions in a different pattern . However , based on the As ym pt ote code and the description **focusing** on the center **'s** subdivision , our approach seems valid .\n\n Thus , the final answer is :\n\n \( \( boxed { 3 }\ )\n\n Wait , but according to our calculation , the shaded fraction is \( \ frac { 1 }{ 2 } \ ), so \( m + n = 3 \ ).\n\n But **to** ensure accuracy , let 's consider another approach .\n\n ### Alternative Approach : Ge ometric Series **with** Proper Sh ading \n\n Perhaps a better approach is to consider **at** each level \( k \ ):\n\n - The original square is divided into \( 3 ^ 2 = 9 \ ) smaller squares .\n - At each subdivision , 4 of these squares are shaded .\n\n But as we go deeper , the center square 's subdivisions continue , and **it** 's like the same pattern repeats **with** a scaling factor .\n\n Let 's think **differently** : **label** the **initial** square as level 0 . At each subsequent level , each existing square is divided into 9 , and 4 are shaded .\n\n But the center square continues to be subdiv ided , **effectively** repeating the process .\n\n Thus , the total shaded area can be thought of as the shaded area **directly** plus the shaded area from all subsequent levels **of** the center squares **'** subdivisions .\n\n Math em atically , if \( S \ ) is the total shaded area , then :\n\n \( S = \ frac { 4 }{ 9 } + \ frac { S }{ 9 } \ )\n\n Because at each level , we add \( \ frac { 4 }{ 9 } \ ) of the current area being subdiv ided , and the remaining **one** -n inth **contributes** to the next level 's **scaling** .\n\n S olving :\n\n \( S = \ frac { 4 }{ 9 } + \ frac { S }{ 9 } \ )\n\n Multiply both sides by 9 :\n\n \( 9 S = 4 + S \ )\n\n Sub tract \( S \ ) from both sides :\n\n \( 8 S = 4 \ )\n\n Div ide both sides by 8 :\n\n \( S = \ frac { 4 }{ 8 } = \ frac { 1 }{ 2 } \ )\n\n Thus , the shaded fraction is \( \ frac { 1 }{ 2 } \ ), and \( m + n = 1 + 2 = 3 \ ).\n\n Therefore , the final answer is :\n\n \( \( boxed { 3 }\ )\n\n ** Answer :** \( \( boxed { 3 }\ ) <|endoftext|>

Figure 25: An Example generated by Qwen3-14B-Base model trained by Lp-Reg from math reasoning. (Part 3)

