# OpenReview forum: "Low-probability Tokens Sustain Exploration in Reinforcement Learning with Verifiable Reward"
_ICLR.cc/2026/Conference — ICLR 2026 Conference Withdrawn Submission_

### Official Review · Reviewer_2YYP · 2025-10-23

**Soundness:** 2
**Presentation:** 3
**Contribution:** 2
**Rating:** 4
**Confidence:** 3

**Summary:**

This paper investigates the exploration dynamics within RLVR and identifies that the gradual elimination of what the authors term "reasoning sparks"—a crucial subset of low-probability tokens  that initiate diverse reasoning paths. While these tokens are abundant in pre-trained models, they are systematically extinguished during RLVR due to over-penalization, leading to a degeneracy in exploration. Previous methods typically address this issue by maintaining high policy entropy, but the precise mechanisms that govern meaningful exploration have remained underexplored. The authors' analysis suggests that an unselective focus on entropy risks amplifying irrelevant tokens and destabilizing training. To address this, the paper introduces Low-probability Regularization, which regularizes the policy towards a heuristic proxy distribution.

**Strengths:**

1.  The motivation that low-probability tokens might indicate valuable exploration is well-argued and supported by analysis.
2.  The authors compare their method with up-to-date baselines.

**Weaknesses:**

1.  The idea of training on low-probability or high-entropy tokens has been heavily investigated in prior work (as cited in the related work section). Compared to these works, the main difference introduced here is the filtering of noisy data, which may make the contribution somewhat limited.
2.  While the authors claim that probability might be a better metric than entropy, the provided support is mainly heuristic; a more theoretical explanation would strengthen the argument.
3.  The authors select different probability percentile thresholds for different base models. It would be helpful to clarify how these hyperparameters are tuned and whether this increases the tuning cost for future applications.

**Questions:**

See weakness.

---

> ### Author Response · Authors · 2025-11-24
> **Response to Reviewer 2YYP (1/3)**
>
> We sincerely thank you for your effort and valuable comments in reviewing our paper. We really appreciate your recognition of our contributions regarding the motivation and experiments, as well as your insightful feedback. In our response, we will address each question individually, quoting them and providing a corresponding answer.
>
> ---
>
> > **Q1 (W1): The idea of training on low-probability or high-entropy tokens has been heavily investigated in prior work (as cited in the related work section). Compared to these works, the main difference introduced here is the filtering of noisy data, which may make the contribution somewhat limited.**
>
> A1: We would like to clarify that our work, which focuses on low-probability tokens, is different and more effective than prior approaches that target high-entropy tokens. We support this claim through three dimensions: empirical results, statistical analysis and theoretical analysis.
>
> (1) **Empirical Superiority: low-probability regularization outperforms high-entropy regularization**:
> *   **Head-to-Head Comparison:** In **Table 2**, our method (Lp-Reg) consistently outperforms the state-of-the-art "high-entropy" method (80/20 Rule) by a significant margin across both 14B and 32B models. This highlights the superior effectiveness of targeting low-probability tokens.
> *   **Controlled Ablation:** In **Appendix B.2 (Figure 10)**, we isolate the variable completely. When we apply our regularization framework to *high-entropy* tokens instead of *low-probability* ones, the method fails to improve performance and suffers from entropy collapse. This confirms that the choice of targeting low-probability tokens is better than high-entropy tokens.
>
> (2) **Statistical Analysis: high-entropy methods miss important low-entropy and low-probability tokens**:
> *   **Word Cloud Statistics:** As shown in the word cloud statistics (**Section 6.1, Figure 5**), high-entropy tokens are predominantly function words ("the," "of") or formatting characters, which carry little exploratory intent. In contrast, low-probability tokens include semantically rich exploratory markers like "wait," "however," and "perhaps," which signal shifts in the reasoning trajectory.
> *   **Probability-Entropy Distribution Statistics:** The probability-entropy plots in **Figure 6** further illustrate that low-probability tokens can be sampled from both high-entropy and low-entropy positions, with a significantly larger proportion originating from low-entropy positions compared to high-entropy ones. Thus, an indiscriminate focus on high-entropy positions would overlook many of these critical tokens.
>
> (3) **Theoretical Analysis: high entropy is a subset of low-probability tokens**:
>
> To provide a rigorous foundation, we have added a theoretical analysis in **Appendix C.1**. We introduce a proposition demonstrating that: When entropy is high enough, the set of tokens targeted by high-entropy methods is a subset of those captured by low-probability methods. Mathematically, high entropy implies low probability (uniform distribution), but low probability does *not* imply high entropy. Therefore, entropy-based methods structurally overlook the "blind spot" of low-probability tokens occurring in low-entropy distributions, which is a region we prove is vital for exploration. (We will expand on this theoretical analysis in our response to Q2).
>
> To our best knowledge, our work is the first to systematically analyze and disentangle these two concepts in the context of RLVR, and our empirical and theoretical analysis provide strong evidence for this comparison.

---

> ### Author Response · Authors · 2025-11-24
> **Response to Reviewer 2YYP (2/3)**
>
> > **Q2 (W2): While the authors claim that probability might be a better metric than entropy, the provided support is mainly heuristic; a more theoretical explanation would strengthen the argument.**
>
> A2: We thank the reviewer for this insightful suggestion. In response, we have added a theoretical explanation in **Appendix C.1** to strengthen our argument.
>
> Our empirical results (**Table 2** and **Figure 10**) have demonstrated the superior performance of regularizing on low-probability tokens rather than high-entropy tokens. To explain this, **we theoretically demonstrate that when entropy is high enough, the set of tokens targeted by high-entropy methods is effectively a subset of those captured by low-probability methods**. Specifically, we establish the following proposition linking policy entropy $\mathcal{H}$ to token probability:
>
> **Proposition 1.** *Given the policy $\pi_{\theta}(\cdot|s)$ over a vocabulary $V$, and the policy entropy defined as $\mathcal{H}(\pi_{\theta}) = -\sum_{o \in V} \pi_{\theta}(o|s) \log(\pi_{\theta}(o|s))$, the following holds:*
>
> $$
> \forall \epsilon \in (1/|V|, 1), \quad \exists \delta > 0, \quad \text{s.t. if } \mathcal{H}(\pi_{\theta}) > \delta, \text{ then } \pi_{\theta}(o|s) < \epsilon, \quad \forall o \in V.
> $$
>
> This proposition (proved in the Appendix C.1) establishes two important points regarding exploration:
>
> 1.  **High Entropy Implies Low Probability:** The existence of high entropy mathematically necessitates that all tokens have low probability (i.e., the distribution is near-uniform). Therefore, methods targeting high entropy are confined to operating within this subset of low-probability tokens.
> 2.  **The Blind Spot of High Entropy (Figure 14):** However, the converse does not hold. Low-probability tokens can also be sampled from low-entropy distributions. This occurs when the model is confident in a dominant token (driving entropy down) but samples a rare alternative.
>
> Entropy-based methods inherently overlook low-probability but low-entropy tokens. Our proposed Lp-Reg, by targeting low probability directly, remains effective across both high-entropy and low-entropy contexts. This theoretical advantage explains why Lp-Reg captures valuable exploratory tokens that entropy-based baselines miss, leading to the superior performance observed in our experiments.

---

> ### Author Response · Authors · 2025-11-24
> **Response to Reviewer 2YYP (3/3)**
>
> > **Q3 (W3): The authors select different probability percentile thresholds for different base models. It would be helpful to clarify how these hyperparameters are tuned and whether this increases the tuning cost for future applications.**
>
> A3: We will address each of your concerns in three parts as follows:
>
> > **(1) Does the hyperparameter tuning increase the cost for future applications?**
>
> We would like to clarify that our method is not sensitive to the probability percentile threshold, $\rho$. **Actually, we have conducted a sensitivity analysis of $\rho$, in Appendix B.4 of our initial ICLR submission.** As shown in **Figure 13(a)**, we evaluated $\rho$ with values of $0.005$, $0.010$, and $0.015$. The results demonstrate that the model's performance remains consistently high across these settings, which indicates the robustness of our approach and mitigates concerns about extensive tuning costs for future applications. To make this clearer, we have now highlighted this sensitivity analysis in the main text where the hyperparameter is introduced.
>
>
> > **(2) Why do you report results of 14B and 32B models using different probability percentile thresholds $\rho$ in the main results?**
>
> The different thresholds reported in our main results for the Qwen3-14B ($\rho=0.01$) and Qwen2.5-32B ($\rho=0.005$) models were a matter of experimental logistics. Given that our comprehensive experiments required approximately 10 days of training on 32 GPUs to complete 1,000 steps, the 1,000 steps training for the 14B model with $\rho=0.005$ had not completely finished by the submission deadline. Since our sensitivity analysis confirms strong performance at both values, we reported the results from the completed runs.
>
> > **(3) Why select values around 0.01 as the initial values of $\rho$?**
>
> Our initial choice for $\rho$ was guided by a data-driven statistical analysis of standard GRPO training dynamics. We have added this guideline to **Appendix B.3** to directly address your concern regarding the tuning cost for future applications.
>
> As detailed in **Appendix B.3, Figure 11**, our analysis of standard GRPO training shows that the policy distribution tends to sharpen over time. We observe that tokens in the lowest 5% percentile can span a wide range up to $0.5$ probability, making them unsuitable candidates for protection. In contrast, the bottom 1% consistently stays below $0.1$, representing the true low-probability tail. Therefore, $\rho \approx 0.01$ emerges as a robust, empirically grounded starting point. Supported by the stability shown in our sensitivity analysis (**Figure 13(a)**), this guideline minimizes the need for future hyperparameter tuning.
>
> ---
>
> **Thank you again for your insightful feedback and constructive suggestions. In addition to the responses above, we have revised the paper accordingly. Please let us know if our responses address your concerns. We sincerely appreciate your time and effort in helping us enhance our work.**

---

> ### Comment · Reviewer_2YYP · 2025-11-26
>
> Thanks for the authors' response; my review remains the same.

---

> > ### Author Response · Authors · 2025-11-26
> > **Response to Reviewer 2YYP**
> >
> > We sincerely thank you for your reply. We have made significant efforts to address your concerns by providing extensive additional clarifications and analyses, including (1) **head-to-head and controlled variable comparisons** showing the empirical superiority of low-probability regularization over entropy-based methods across diverse model scales (from 8B to 32B) and domains (math, code, science), (2) **theoretical analysis and statistical evidence** distinguishing low-probability tokens from high-entropy ones, and (3) **a comprehensive sensitivity analysis** demonstrating the robustness of our hyperparameters. **Could you please kindly let us know if these specific responses and data address your concerns?** We truly value your feedback and hope to engage in a constructive discussion to clarify any remaining doubts.
> >
> > We fully acknowledge that entropy-based methods have made significant contributions to the community by mitigating exploration collapse in the early stages of training. Building on these foundations, our work aims to address a critical bottleneck in **long-horizon RL scaling**: sustaining exploration beyond the initial phase to thousands of training steps. Supported by a cumulative total of **300,000 GPU-Hours** of experiments, we identified that targeting low-probability tokens is more effective for scalability than entropy-based methods. We validated this by achieving stable on-policy training for **3,000 steps** (consuming 81,204 GPU-Hours for a single run) without performance collapse, a regime where many baseline methods struggle.
> >
> > Our goal is to contribute these principled insights and large-scale empirical experiences to the community to facilitate the scaling of reasoning models. To this end, we have committed to publicly releasing our code and training scripts. We remain open to further discussion on the scientific nuances of exploration dynamics and sincerely hope our extensive analysis provides a valuable perspective for the field.

---

### Official Review · Reviewer_CCqW · 2025-10-26

**Soundness:** 3
**Presentation:** 3
**Contribution:** 2
**Rating:** 4
**Confidence:** 4

**Summary:**

This work identifies that exploration collapse in Reinforcement Learning with Verifiable Rewards (RLVR)  might be caused by the loss of valuable, low-probability reasoning actions. To counter this, the authors propose Low-probability Regularization (Lp-Reg), a method that distinguishes meaningful low-probability tokens from noise by leveraging their relatively higher likelihood in local predictive contexts.

By filtering out noise and regularizing the policy toward these high-value sparks, Lp-Reg preserves the policy’s useful low-probability tail, enabling more stable training and achieving about 2.66% improvement in test accuracy over baselines.

**Strengths:**

(1) This paper is well-written
(2) The visualizations are clear and rich in information
(3) The design method is simple and effective, and has good generalization potential.
(4)The exploration issues that this article focuses on are indeed the key problems recognized in the current RLVR field.
(5) The baselines in the experiment were all relatively new
(6) The correlation between low-probability tokens and entropy was explored for the first time

**Weaknesses:**

- Expert dependence in hyperparameter configuration: The method’s parameter settings rely heavily on expert prior knowledge, which may hinder its accessibility and ease of use for practitioners without domain-specific experience.

- Computational overhead of the regularization procedure: The proposed regularization appears to involve, for each training batch, a full-vocabulary statistical analysis and ranking of token logits. Given the size of modern language model vocabularies (often >50K tokens), this step could incur prohibitive computational costs, raising concerns about scalability.

- Need for trajectory-level analysis of low-probability tokens: The current analysis lacks a trajectory-level characterization of low-probability tokens. Under Monte Carlo sampling, advantages within a single trajectory are typically uniformly positive or negative. It would be highly informative to examine how low-probability tokens are distributed across positive versus negative trajectory segments—this could reveal whether such tokens are genuinely associated with high-quality reasoning or merely stochastic noise.

- Clarification of conceptual distinction from prior work: A related study [1] also investigates the role of low-probability tokens in policy learning but appears to reach a contrasting conclusion. The authors are encouraged to provide a detailed comparative analysis that clearly articulates the fundamental differences in assumptions, definitions (e.g., what constitutes a “valuable” low-probability token), and methodological approaches between this work and [1]. Highlighting this distinction will strengthen the novelty and conceptual contribution of the current paper.

[1]: Do Not Let Low-Probability Tokens Over-Dominate in RL for LLMs

**Questions:**

Please refute the above-mentioned weakness through experiments or discussions. I will adjust the score based on the author's feedback and the opinions of other reviewers

---

> ### Author Response · Authors · 2025-11-24
> **Response to Reviewer CCqW (1/5)**
>
> We sincerely thank you for your effort and valuable comments in reviewing our paper. We greatly appreciate your insightful feedback and constructive questions. We also appreciate your recognition of our contributions in our presentation, method, experiments and analysis. In our response, we will address each question individually, quoting them and providing a corresponding answer.
>
> ---
>
> > **Q1 (W1): Expert dependence in hyperparameter configuration: The method’s parameter settings rely heavily on expert prior knowledge, which may hinder its accessibility and ease of use for practitioners without domain-specific experience.**
>
> A1: We would like to address this concern through three aspects: **(1) robust sensitivity analysis**, **(2) explicit data-driven selection guidelines**, and **(3) extensive generalization experiments using a unified hyperparameter set**.
>
> **(1) Robust Sensitivity Analysis**
> We would like to clarify that our initial submission included a comprehensive sensitivity analysis in **Appendix B.4** for the two core hyperparameters: the low-probability percentile $\rho$ and the min-p ratio $\kappa$ (which determines the noise threshold $\tau$). Extensive 1,000 training steps' long-horizon training experiments (spanning approx. 48,000 GPU hours) demonstrate that model performance remains consistently high across a reasonable range of values. This stability indicates the intrinsic robustness of our approach and mitigates concerns regarding tuning costs for future applications.
>
> **(2) Data-Driven Guidelines for Value Selection**
> For further accessibility, we have added **Appendix B.3**, which provides clear, data-driven guidelines for parameter selection based on the intrinsic training dynamics of a standard GRPO baseline. This allows practitioners to identify rational values without heuristic guessing:
> *   **Selection of $\rho$:** As visualized in **Figure 11**, our analysis of standard GRPO training reveals a clear distinction in the low-probability tail. While tokens in the bottom 5% can reach probabilities as high as $0.5$ (due to probability concentration), tokens in the bottom 1% consistently remain in the strictly low-probability regime ($< 0.1$). Consequently, setting $\rho \approx$ 1% ($0.01$) is a logical, empirically derived choice to target the true tail without inadvertently regularizing high-probability tokens.
> *   **Selection of $\kappa$:** As shown in **Figure 12**, we identify a persistent gap in the relative probability ratios between meaningful reasoning sparks and meaningless noise tokens. Noise tokens typically fall below a ratio of $0.01$, whereas exploratory tokens consistently remain above it. This empirical observation justifies setting $\kappa \approx 0.01$ as an effective initial value to filter noise while preserving reasoning capabilities.
>
> **(3) Generalization Across Models and Domains**
>
> To definitively validate the accessibility and ease of use, we extended our experiments to different architectures (Llama3) and domains (Science and Code) in **Appendix B.6** of the revision. As shown in **Table 5** and **Table 6**, Lp-Reg achieves significantly better performance than baselines using the **exact same hyperparameters** across all settings. Notably, we demonstrated exceptional long-term stability by training the Qwen2.5-32B base model for 3,000 steps, consuming 81,204 GPU hours as shown in **Figure 2**, as part of a comprehensive experimental campaign totaling over 300,000 GPU hours. The fact that the same configuration works effectively across various model sizes (8B, 14B, 32B), model families (Qwen, Llama), and domains (Math, Code, Science) strongly proves that Lp-Reg has great potential for long-horizon RL scaling and does not require domain-specific expertise for hyperparameter tuning.

---

> ### Author Response · Authors · 2025-11-24
> **Response to Reviewer CCqW (2/5)**
>
> > **Q2 (W2): Computational overhead of the regularization procedure: The proposed regularization appears to involve, for each training batch, a full-vocabulary statistical analysis and ranking of token logits. Given the size of modern language model vocabularies (often >50K tokens), this step could incur prohibitive computational costs, raising concerns about scalability.**
>
> A2: We would like to clarify a potential misunderstanding regarding the "full-vocabulary ranking" and provide evidence that the cost is negligible. We have added **Appendix B.7 (Computational Overhead Analysis)** to elaborate on this.
>
> **(1) Clarification on Algorithmic Complexity**
>
> We clarify that **Lp-Reg does NOT require sorting or ranking the full vocabulary for each token.** The process consists of two parts, neither of which incurs prohibitive costs:
> *   **Renormalization (Proxy Construction, Equation 5 in pdf):** As shown in Listing 1 in the revised pdf, this step only requires calculating the "max" and "exp" operations over the vocabulary dimension. Its complexity is $\mathcal{O}(|V|)$, which is **identical to the standard Softmax/Log-Softmax operations** already inherently required by the LLM's loss function. It uses efficient, parallelizable vector operations on the GPU and does not involve sorting.
> *   **Regularization (Loss computation, Equation 6 in pdf):** As shown in Listing 2 in the revised pdf, The sorting/ranking is performed **globally on the micro-batch** to find the bottom $\rho$% tokens, *not* on the vocabulary dimension for every position. The complexity depends on the number of tokens in the batch ($N$), i.e., $\mathcal{O}(N)$ or $\mathcal{O}(N \log K)$, which is independent of the vocabulary size $|V|$.
>
> **(2) Empirical Verification**
>
> To validate this experimentally, we measured the actual training time overhead in the following **Table R1**. Even with a standard vocabulary size (e.g., Qwen3's vocabularies around 150k), Lp-Reg adds only 0.30% $\sim$ 0.50% extra time per training step compared to the standard GRPO. This confirms that the operations are highly efficient and do not become a bottleneck, ensuring scalability even for models with large vocabularies.
>
> **Table R1 (Table 7 in pdf):** Runtime comparison between GRPO and Lp-Reg under different training steps. Lp-Reg introduces only marginal overhead compared with GRPO.
> |Steps|Avg. Response Length|GRPO (s)|Lp-Reg (s)|Overhead|
> |:---:|:---:|:---:|:---:|:---:|
> |256|4058.53|698.49|700.60|+0.30%|
> |512|5794.25|973.74|978.62|+0.50%|
> |768|6640.69|1137.24|1141.49|+0.37%|

---

> ### Author Response · Authors · 2025-11-24
> **Response to Reviewer CCqW (3/5)**
>
> > **Q3 (W3): Need for trajectory-level analysis of low-probability tokens: The current analysis lacks a trajectory-level characterization of low-probability tokens. Under Monte Carlo sampling, advantages within a single trajectory are typically uniformly positive or negative. It would be highly informative to examine how low-probability tokens are distributed across positive versus negative trajectory segments—this could reveal whether such tokens are genuinely associated with high-quality reasoning or merely stochastic noise.**
>
> A3: To address this concern, we have added a **Trajectory-Level Token Analysis** in **Appendix C.2** in the revised submission, where we decomposed token distributions into positive ($A>0$), negative ($A<0$), and invalid ($A=0$) trajectories. Our findings strongly refute the hypothesis that low-probability tokens are merely stochastic noise:
>
> **(1) Low-probability exploratory tokens are reasoning patterns**
>
> As visualized in **Figure 16**, the probability distributions of exploratory tokens (e.g., "wait", "but") are similar across positive and negative trajectories both on standard GRPO and Lp-Reg. This indicates that these tokens function as reasoning patterns: they represent the mechanism of a reasoning attempt, rather than the determinant of the final outcome. To use an analogy provided in our revision: Just as scratchpad paper is utilized for both correct and incorrect math solutions, a negative trajectory containing "wait" represents a failed reasoning attempt. This is different from a failure due to a lack of reasoning which baselines suffer from.
>
> **(2) Low-probability exploratory tokens indicate active exploration**
>
> We observed a sharp contrast in sampling density between active learning groups (where $A \neq 0$, implying mixed success/failure) and static groups ($A=0$). Exploratory tokens appear with significantly higher density in the active groups. This confirms that these tokens are most active when the model is exploring diverse outcomes.
> However, because these tokens naturally appear abundantly in negative trajectories (simply due to the high volume of failed attempts during exploration), standard GRPO treats them as "bad actions" and systematically suppresses them.
>
> **(3) Low-probability exploratory tokens' sampling is correlated with performance collapse**
>
> Crucially, our analysis reveals the cause of the baseline's failure. As shown in **Figure 17** (Step 512), standard GRPO exhibits a significant reduction in sampling these low-probability tokens in later stages, which directly correlates with the performance bottleneck (plateau) observed in **Figure 17**. In contrast, Lp-Reg maintains robust sampling of these tokens throughout long-horizon training, coinciding with continuously increasing accuracy. This demonstrates that preserving these low probability tokens in negative trajectories is effective to sustain exploration and performance.

---

> ### Author Response · Authors · 2025-11-24
> **Response to Reviewer CCqW (4/5)**
>
> > **Q4 (W4): Clarification of conceptual distinction from prior work: A related study [1] also investigates the role of low-probability tokens in policy learning but appears to reach a contrasting conclusion. The authors are encouraged to provide a detailed comparative analysis that clearly articulates the fundamental differences in assumptions, definitions (e.g., what constitutes a “valuable” low-probability token), and methodological approaches between this work and [1]. Highlighting this distinction will strengthen the novelty and conceptual contribution of the current paper.**
>
> [1]: Do Not Let Low-Probability Tokens Over-Dominate in RL for LLMs
>
> A4: We would like to clarify that our work (Lp-Reg) and Lopti[1] are not in conflict; rather, they identify and address two distinct orthogonal challenges in RLVR training. Lopti focuses on improving gradient dynamics for better **data efficiency**, while our work focuses on ensuring **long-term training stability**. We will elaborate on this distinction from three perspectives: the core research problem, the methodological approach, and new, direct experimental comparisons.
>
> **(1) Different Core Research Problems**
>
> Lopti focuses on the **training efficiency**, whereas Lp-Reg focuses on the **training stability**. These represent two orthogonal axes of improvement for RLVR.
>
> - Lopti identifies that low-probability tokens generate gradients with disproportionately large norms. Its core focus is on how this "over-domination" suppresses gradient signals from high-probability tokens, thereby reducing the data efficiency of the training process.
>
> - In contrast, our Lp-Reg identifies the systematic elimination of low-probability tokens with exploratory semantics (e.g., "wait"), which we term "reasoning sparks." Our core focus is on how the over-penalization of these tokens leads to a loss of exploration capacity with the entropy collapse phenomenon, hindering the model from achieving higher performance in long-horizon stable training.
>
> **(2) Different Methodological Approaches**
>
> Lopti's method of **separate gradient updates** and Lp-Reg's **regularization** are distinct and non-conflicting algorithms.
>
> - To prevent large-norm gradients from low-probability tokens suppressing signals from high-probability tokens, Lopti separates the loss computation for these two groups and updates the model parameters twice per micro-batch.
>
> - To protect low-probability tokens from over-penalization in RLVR, Lp-Reg introduces a regularization on them via a KL divergence between the current policy and a filtered proxy policy.
>
> **(3) Empirical evidence from long-horizon experiments**
>
> To empirically validate our claims, we have conducted a long-horizon training experiment comparing Lopti, Lp-Reg, and the GRPO baseline for 1,000 steps. Results shown in **Figure 18**  and the following **Table R2** clearly illustrate their different effects.
>
> **Table R2 (corresponding to Figure 18 in pdf)**: Performance comparison of on-policy GRPO, Lopti, and Lp-Reg on the Qwen3-14B-Base model.  Best scores are **bolded**.
> |Methods|AIME24|AIME25|Math-500|Minerva|Olympiad Bench|avg.|
> |:---|---:|---:|---:|---:|---:|---:|
> |GRPO (on.)|46.04|34.38|93.00|48.53|65.19|57.43|
> |Lopti (on.)|46.25|31.67|93.00|**50.74**|65.33|57.40|
> |Lp-Reg (on.)|**50.83**|**37.92**|**94.40**|49.26|**68.44**|**60.17**|
>
> - **Lopti excels in data efficiency**: The Lopti curve (orange) shows a faster initial rise in test accuracy, confirming its effectiveness at accelerating learning, consistent with the findings in their paper.
>
> - **Lopti fails to sustain exploration in long-horizon RLVR**: After approximately 600 steps, Lopti's performance plateaus, and its training entropy collapses in the same manner as the GRPO baseline. This shows that improving data efficiency does not inherently solve the long-term exploration problem.
>
> - **Lp-Reg excels in sustained exploration**: In contrast, our Lp-Reg (purple) demonstrates stable performance improvement throughout the 1,000 steps, correlated with its ability to maintain policy entropy. This sustained exploration allows it to achieve a significantly higher final accuracy.
>
> In conclusion, Lp-Reg and the Lopti study address distinct, orthogonal challenges in RLVR. The choice between these methods may depend on the specific training objectives. While investigating a potential combination could be an interesting avenue for future research, our primary contribution here is to formally identify the exploration stability and provide an effective solution for it. We have added this detailed comparison to our revised manuscript to better contextualize our work and highlight its unique conceptual contribution.
> [1]: Do Not Let Low-Probability Tokens Over-Dominate in RL for LLMs

---

> ### Author Response · Authors · 2025-11-24
> **Response to Reviewer CCqW (5/5)**
>
> > **Q5 (Q1): Please refute the above-mentioned weakness through experiments or discussions. I will adjust the score based on the author's feedback and the opinions of other reviewers.**
>
> A5: Thank you again for your insightful feedback and constructive suggestions. We are pleased to address your concerns (W1–W4) through the detailed responses above and the corresponding revisions in our manuscript. Specifically, we have demonstrated robust hyperparameters via data-driven guidelines and sensitivity analysis (refuting W1), verified negligible computational overhead ($< 0.5%$) (refuting W2), provided trajectory-level evidence that low-probability tokens represent consistent reasoning patterns rather than noise (refuting W3), and empirically distinguished Lp-Reg’s focus on long-term stability from prior work's focus on data efficiency (refuting W4).
>
> ---
>
> **In addition to the responses above, we have revised the paper accordingly. Please let us know if our responses address your concerns. We sincerely appreciate your time and effort in helping us enhance our work.**

---

### Official Review · Reviewer_9MFy · 2025-10-27

**Soundness:** 2
**Presentation:** 3
**Contribution:** 2
**Rating:** 4
**Confidence:** 4

**Summary:**

This paper studies why RLVR often stops improving when models lose exploration ability. The authors find that during RLVR training, certain rare but important reasoning sparks gradually disappear because they are over-penalized. To fix this, they propose Lp-Reg, which protects these exploratory tokens by regularizing the policy toward a filtered, re-normalized proxy distribution. This approach stabilizes training and maintains exploration for longer, leading to better reasoning performance, with a 60.17% average accuracy on five math benchmarks, outperforming previous methods by 2.66%.

**Strengths:**

1. Novel insight into exploration collapse: this work introduces the concept of ***reasoning sparks***, offering a fresh perspective on why exploration diminishes during RLVR training.


2. Simple yet effective method: the proposed Lp-Reg is lightweight and easy to implement, improving exploration stability without architectural changes.


3. Strong empirical performance: this work achieves state-of-the-art results on five math reasoning benchmarks, outperforming previous methods by 2.66% on average.


4. Improved training stability: Lp-Reg effectively prevents early entropy collapse, enabling longer and more stable on-policy training phases.

**Weaknesses:**

1. Heuristic dependency: the proposed method relies on manually selected probability thresholds and re-normalization heuristics, lacking strong theoretical justification. As shown in Eq. 6, additional hyperparameters such as $\tau$ and $\delta_\rho$ are introduced, which require extra tuning and may increase the method’s sensitivity to hyperparameter selection.


2. Incomplete mechanistic understanding: while the paper identifies the disappearance of ***reasoning sparks***, the precise connection between these tokens and the underlying reasoning structures remains ambiguous. The authors state that “while both are empirically low-probability tokens within a long trajectory, a reasoning spark often has a higher relative probability than a noise token in the immediate next-token prediction.” Please provide quantitative evidence to support this claim, and clearly define what constitutes a reasoning spark to distinguish it from random low-probability tokens.


3. Potential computational cost: the additional filtering and redistribution of token probabilities may introduce non-trivial computational overhead, especially for large-scale models. Please include an analysis or estimation of the extra computational complexity incurred by Lp-Reg, and discuss whether it affects training efficiency or scalability.

**Questions:**

Q1. In Figure 1 (c) and (d), what does the variable $n$ represent?


Q2. When comparing GRPO in Eq. 3 and Lp-Reg in Eq. 6, the normalization strategies differ: the former uses $\frac{1}{G} \sum_{i=1}^{G} \frac{1}{|o_i|} \sum_{t=1}^{|o_i|}$ while the latter adopts $\frac{1}{\sum_{i=1}^{G} |o_i|} \sum_{i=1}^{G} \sum_{t=1}^{|o_i|}$. As far as I know, the latter formulation provides a more accurate normalization over all tokens. Could the reported performance improvement be partially attributed to this modification rather than the proposed regularization itself?

---

> ### Author Response · Authors · 2025-11-24
> **Response to Reviewer 9MFy (1/3)**
>
> We sincerely thank you for your effort in reviewing our paper. We greatly appreciate your valuable comments and insightful feedback. We also appreciate your recognition of our contributions in our novel insight, simple yet effective method, strong empirical performance, and improved training stability. In our response, we will address each question individually, quoting them and providing a corresponding answer.
>
> ---
>
> > **Q1 (W1): Heuristic dependency: the proposed method relies on manually selected probability thresholds and re-normalization heuristics, lacking strong theoretical justification. As shown in Eq. 6, additional hyperparameters such as
> $\tau$ and $\delta_{\rho}$ are introduced, which require extra tuning and may increase the method’s sensitivity to hyperparameter selection.**
>
> A1: We would like to clarify that our approach is designed to eliminate heuristic guessing through **robust sensitivity analysis**, **explicit data-driven derivation**, and **universal cross-domain generalization**.
>
> **(1) Robust Sensitivity Analysis**
>
> First, to demonstrate that our method is not brittle and does not require expert-level fine-tuning, we conducted a comprehensive sensitivity analysis in **Appendix B.4**. Through extensive long-horizon training experiments (spanning approximately 48,000 GPU hours), we verified that Lp-Reg maintains consistently high performance across a reasonable range of values for both the low-probability percentile $\rho$ and the min-p ratio $\kappa$. This inherent stability implies that practitioners do not need to precisely tune these parameters to achieve optimal results, significantly lowering the risk of sensitivity.
>
> **(2) Elimination of Heuristics via Data-Driven Guidelines**
>
> To directly address the concern about "heuristic dependency," we have added **Appendix B.3**, which provides a rigorous, data-driven guideline for parameter selection based on the training dynamics of the standard GRPO baseline. This allows users to identify rational values through statistical observation rather than guessing:
>
> *   **Selection of $\rho$:** As visualized in **Figure 11**, analyzing the probability distribution of the baseline reveals a sharp boundary. Tokens in the bottom 5% can reach probabilities as high as $0.5$, whereas the bottom 1% consistently remains in the strictly low-probability regime ($< 0.1$). Thus, setting $\rho \approx 0.01$ is not a heuristic guess but a logical choice derived from the data distribution to target the true low-probability tokens rather than high-probability tokens.
> *   **Selection of $\kappa$:** As shown in **Figure 12**, we identify a distinct statistical gap in the relative probability ratios. There is a persistent empirical margin in low-probability region where meaningful exploratory tokens (e.g. wait) consistently score higher than meaningless noise (e.g. button). This empirical observation provides an objective standard for setting $\kappa \approx 0.01$, effectively filtering noise without requiring domain-specific prior knowledge.
>
> **(3) Universal Applicability Across Models and Domains**
>
> To address the concern about expert dependence, in **Appendix B.6**, we extended our evaluation to diverse model architectures (Llama3) and distinct domains (Science and Code).
>
> Crucially, as shown in **Table 5** and **Table 6**, Lp-Reg achieves superior performance using the **exact same hyperparameters** across all these settings. Notably, we demonstrated exceptional long-term stability by training the Qwen2.5-32B base model for 3,000 steps (accruing 81,204 GPU hours, as shown in **Figure 2**), as part of our broader experimental campaign totaling over 300,000 GPU hours. The fact that a single hyperparameter configuration generalizes effectively across different model sizes (8B, 14B, 32B), model families (Qwen, Llama), and domains (Math, Code, Science) definitively proves that **Lp-Reg is accessible and easy to use without the need for domain-specific expertise.**

---

> ### Author Response · Authors · 2025-11-24
> **Response to Reviewer 9MFy (2/3)**
>
> > **Q2 (W2): Incomplete mechanistic understanding: while the paper identifies the disappearance of reasoning sparks, the precise connection between these tokens and the underlying reasoning structures remains ambiguous. The authors state that “while both are empirically low-probability tokens within a long trajectory, a reasoning spark often has a higher relative probability than a noise token in the immediate next-token prediction.” Please provide quantitative evidence to support this claim, and clearly define what constitutes a reasoning spark to distinguish it from random low-probability tokens.**
>
> A2: As described in the Introduction, **"reasoning sparks" are empirically identified as low-probability exploratory tokens** such as "wait," "however," or "perhaps." 'Low-probability' describes their statistical attribute, while 'exploratory' describes their semantic function to start a new exploration fork. Thus, a reasoning spark is not a random low-probability token; it is a member of a semantically coherent class that also exhibits a specific statistical behavior. In our analysis, we examine the statistical characteristics of tokens like "but," "wait," and other commonly recognized exploratory tokens, as these are frequently analyzed as representative cases in previous studies [1,2,3,4,5].
>
> For the quantitative evidence, we analyze the next-token prediction distribution throughout the training process and add the analysis to **Section 6.3** in the revised paper. We focused on the top-64 predicted tokens, examining only those with a probability below 0.1 to isolate the low-probability regime. We then tracked the average next-token probability for two distinct token classes:
> - **Reasoning Sparks:** A set of meaningful exploratory tokens ("wait," "perhaps," etc.).
> - **Irrelevant Noise:** A set of semantically irrelevant tokens ("cost," "fine," etc.).
>
> The results, presented below in **Table R1**, reveal a statistically significant and persistent gap.
>
> **Table R1 (values from Figure 8 in pdf):** Probabilistic distinction between reasoning sparks and irrelevant tokens across training steps. The average next-token prediction probability for reasoning sparks is consistently ~2x higher than for noise tokens.
> |Table A-1|Step 64|Step 128|Step 192|Step 256|Step 320|Step 384|
> |-------------------|:-------:|:--------:|:--------:|:--------:|:--------:|:--------:|
> |**Reasoning Sparks**|0.0060|0.0071|0.0074|0.0077|0.0084|0.0085|
> |**Irrelevant Noise**|0.0030|0.0035|0.0028|0.0031|0.0040|0.0034|
>
>
> This data provides strong quantitative support for our claim: across all training stages, reasoning sparks consistently have a higher relative probability than noise tokens. This phenomenon can be attributed to the model's intrinsic confidence [6,7,8]; even when uncertain, the model is 'more confident' in a potential reasoning spark than in random noise.
>
>
> It suggests that while a perfect separation is not possible, a probability threshold $\tau$, as defined for our proxy distribution in **Section 6.3**, can serve as a principled filtering mechanism. Additional sensitivity analysis presented in **Appendix B.4** further highlight the robustness of $\tau$ across different values. It allows us to screen out a large fraction of the low-confidence noise while preserving the majority of the higher-confidence (though still low-probability) reasoning sparks. This resolves the ambiguity and clarifies how Lp-Reg selectively and robustly preserves high-quality exploration. We have included this analysis in Section 6.3 (Figure 8) of our revised manuscript.
>
>
> For greater clarity, we have refined the statement in the Introduction of the revised paper to explicitly contrast these types. In detail, we replace the state *"while both are empirically low-probability tokens within a long trajectory, a reasoning spark often has a higher relative probability than a noise token in the immediate next-token prediction."* to *"While both are empirically low-probability tokens, a meaningful exploratory token like "wait" often has a higher relative probability than a noise token like "cost" in the immediate next-token prediction. "*. And add the support Appendix subsequently.
>
> [1] DeepSeek-R1: Incentivizing Reasoning Capability in LLMs via Reinforcement Learning
>
> [2] Muennighoff et al. S1: Simple test-time scaling
>
> [3] Hu et al. Why Distillation can Outperform Zero-RL: The Role of Flexible Reasoning
>
> [4] Qian et al. Demystifying Reasoning Dynamics with Mutual Information: Thinking Tokens are Information Peaks in LLM Reasoning
>
> [5] Wang et al. Beyond the 80/20 Rule: High-Entropy Minority Tokens Drive Effective Reinforcement Learning for LLM Reasoning
>
> [6] Nguyen et al. Turning up the heat: Min-p sampling for creative and coherent llm outputs.
>
> [7] Xu et al. Adaptive termination for multi-round parallel reasoning: An universal semantic entropy-guided framework.
>
> [8] Fu et al. Deep Think with Confidence.

---

> ### Author Response · Authors · 2025-11-24
> **Response to Reviewer 9MFy (3/3)**
>
> > **Q3 (W3): Potential computational cost: the additional filtering and redistribution of token probabilities may introduce non-trivial computational overhead, especially for large-scale models. Please include an analysis or estimation of the extra computational complexity incurred by Lp-Reg, and discuss whether it affects training efficiency or scalability.**
>
>
> A3: We have included a detailed **computational overhead analysis** in **Appendix B.7** to address this. In the analysis, we conducted empirical runtime comparison and theoretical complexity analysis.
>
> **(1) Empirical Runtime Analysis (< 0.5% Overhead)**
>
> To strictly quantify the cost, we conducted a controlled runtime comparison between the standard GRPO and Lp-Reg. As shown in **Table R2** (Table 7 in pdf) below, we measured the execution time of a single training step under identical conditions (same checkpoint, random seed, and response lengths) at various training stages (steps 256, 512, 768).
> The results indicate that Lp-Reg introduces a **negligible relative overhead of approximately 0.30% $\sim$ 0.50%**. For example, at step 512 with an average response length of ~5.8k tokens, the step time increased marginally from 973.74s (GRPO) to 978.62s (Lp-Reg).
>
> **Table R2 (Table 7 in pdf):** Runtime comparison between GRPO and Lp-Reg under different training steps. Lp-Reg introduces only marginal overhead compared with GRPO.
> |Steps|Avg. Response Length|GRPO (s)|Lp-Reg (s)|Overhead|
> |:---:|:---:|:---:|:---:|:---:|
> |256|4058.53|698.49|700.60|+0.30%|
> |512|5794.25|973.74|978.62|+0.50%|
> |768|6640.69|1137.24|1141.49|+0.37%|
>
> **(2) Complexity Analysis**
>
> Theoretically, the overhead is minimal because the operations added by Lp-Reg are computationally cheap compared to the Transformer's backbone calculations:
> - **Renormalization for proxy policy (Equation 5 in pdf):** As shown in Listing 1 in the revised pdf, this step involves finding the maximum value and masking, which scales linearly with vocabulary size $\mathcal{O}(|V|)$. However, this is structurally identical to the standard Softmax operation, which is memory-bandwidth bound rather than compute-bound.
> - **Regularization in loss computation (Equation 6 in pdf):** As shown in Listing 2 in the revised pdf, the sorting operation (Top-K) is performed on the flattened batch of tokens ($N \approx 30k$), not on the vocabulary dimension for every token. The complexity is $\mathcal{O}(N)$, which is trivial compared to the $\mathcal{O}(d_{model}^2)$ matrix multiplications in the model.
>
> Thus, Lp-Reg is computationally lightweight and does not negatively affect the training efficiency or scalability of large-scale models.
>
> ---
>
> > **Q4 (Q1): In Figure 1 (c) and (d), what does the variable $n$ represent?**
>
> A4: In Figure 1 (c) and (d), "n" represents the total count of sampled tokens for each distribution. We have added this explanation to the figure caption in our revised paper for better clarification.
>
> ---
>
> > **Q5 (Q2): When comparing GRPO in Eq. 3 and Lp-Reg in Eq. 6, the normalization strategies differ: the former uses $\frac{1}{G} \sum_{i=1}^G \frac{1}{|o_i|} \sum_{t=1}^{|o_i|}$ while the latter adopts $\frac{1}{\sum_{i=1}^G|o_i|} \sum_{i=1}^G \sum_{t=1}^{|o_i|}$. As far as I know, the latter formulation provides a more accurate normalization over all tokens. Could the reported performance improvement be partially attributed to this modification rather than the proposed regularization itself?**
>
> A5: We would like to clarify that the formula in Equation 3 was written to match the formulation in the original GRPO paper[9]. However, for our actual experiments, we adopted the more stable 'token-mean' loss aggregation recipe ($\frac{1}{\sum_{i=1}^G|o_i|} \sum_{i=1}^G \sum_{t=1}^{|o_i|}$) for **all** compared methods, including the standard GRPO baseline and our Lp-Reg. This approach follows the implementation in DAPO [10] to ensure a fair and stable comparison.
>
> Therefore, the observed performance improvement is indeed attributable to our proposed regularization and not to a difference in the loss aggregation scheme. For clarity and consistency, we have modified Equation 3 in our revised manuscript to be $\frac{1}{\sum_{i=1}^G|o_i|} \sum_{i=1}^G \sum_{t=1}^{|o_i|}$, ensuring that all equations now align perfectly with the experimental implementation.
>
> [9] Shao et al. DeepSeekMath: Pushing the Limits of Mathematical Reasoning in Open Language Models.
>
> [10] Yu et al. DAPO: An Open-Source LLM Reinforcement Learning System at Scale.
>
> ---
> **Thank you again for your insightful feedback and constructive suggestions. In addition to the responses above, we have revised the paper accordingly. Please let us know if our responses address your concerns. We sincerely appreciate your time and effort in helping us enhance our work.**

---

> > ### Comment · Reviewer_9MFy · 2025-11-25
> > **Rating update**
> >
> > Thanks to the authors for their response. My concerns regarding the “Reasoning Sparks”, the computational cost, and the inconsistency between Eq. 3 and Eq. 6 have been addressed. I will update my score accordingly.

---

> ### Author Response · Authors · 2025-11-25
> **Thank you for the positive re-evaluation**
>
> We deeply appreciate your thoughtful feedback and are especially grateful for the improved score, which is truly encouraging.  Your constructive feedback helped us enhance our work, and we have certainly integrated your suggestions into our revised version.

---

### Official Review · Reviewer_2rNy · 2025-11-06

**Soundness:** 3
**Presentation:** 2
**Contribution:** 2
**Rating:** 4
**Confidence:** 3

**Summary:**

They first reveal that the existing methods that have an unselective focus on entropy risks amplifying irrelevant tokens and destabilizing training. To address this, they propose Lp-Reg, which applies a probability threshold to filter out noise tokens and then re-normalizes the distribution over the remaining candidates. They compare their method with other training methods and show a high stability. Their method achieves higher performances than others based on five math benchmarks.

**Strengths:**

- They introduce a new training scheme, Lp-Reg, a method that creates a more stable exploratory environment by filtering out presumed meaningless noise to protect the remaining low-probability tokens.
- They show that Lp-Reg achieves state-of-the-art performance with five math benchmarks.
- Their methodology is principled and simple.
- They also conduct ablation studies for robustness.

**Weaknesses:**

Even though they claim that their methodology achieves a higher stability and accuracy than other existing methodologies, the current experiments are limited to the math domain, and the results are not consistent across all benchmarks (i.e., their method is not always the best). Because the reported improvement numbers themselves look small, it is unclear whether the differences are statistically significant. Moreover, their domain is only math, and they evaluate only models trained with one math dataset. This raises concerns about the robustness of the results. For example, if the models were trained on different math datasets, would the results remain the same? Similarly, would the same result appear in other domains such as coding? The tested model family is also limited to Qwen, which further questions the robustness.

I'm also wondering whether the proposed technique would eventually show a performance collapse after 1,000 training steps. Reporting the point at which their method begins to collapse (if it does) could be useful and interesting.

**Questions:**

Please see the weaknesses

---

> ### Author Response · Authors · 2025-11-24
> **Response to Reviewer 2rNy (1/5)**
>
> We sincerely thank you for your effort and valuable comments in reviewing our paper. We really appreciate your recognition of our contributions and your insightful feedback. We also appreciate your recognition of our contributions in our method, performance and ablation study. In our response, we will address each question individually, quoting them and providing a corresponding answer.
>
> ---
>
> > **Q1 (W1) Even though they claim that their methodology achieves a higher stability and accuracy than other existing methodologies, the current experiments are limited to the math domain, and the results are not consistent across all benchmarks (i.e., their method is not always the best). Because the reported improvement numbers themselves look small, it is unclear whether the differences are statistically significant. Moreover, their domain is only math, and they evaluate only models trained with one math dataset. This raises concerns about the robustness of the results. For example, if the models were trained on different math datasets, would the results remain the same? Similarly, would the same result appear in other domains such as coding? The tested model family is also limited to Qwen, which further questions the robustness.**
>
>
> We would like to address your concerns about (1) "not always the best", (2) small improvement, (3) robustness on other domains and model families, respectively.
>
> > **Q-1.1 The results are not consistent across all benchmarks (i.e., their method is not always the best).**
>
> A-1.1: We would like to clarify the evaluation protocol for our Main Results in **Table 2** (copied below). To ensure a rigorous and fair comparison, all scores reported for a given method in a single row are derived from the **single checkpoint** that achieved the highest **average accuracy** across all five benchmarks.
> While this approach reflects the model's holistic reasoning capability, it inevitably involves trade-offs; the checkpoint with the best *average* performance may not correspond to the *peak* performance for every individual benchmark. Therefore, the **Average** score is the most important metric for assessing overall robustness in this table. As shown in **Table 2**, Lp-Reg achieves the highest average accuracy on both the Qwen2.5-32B and Qwen3-14B models. Although Lp-Reg may not top every sub-benchmark at this specific checkpoint, it consistently ranks as either the best or a very close second, demonstrating superior generalist reasoning ability.
>
> **Table 2 copied from pdf:** Main results on five mathematical reasoning benchmarks. On-policy (on.) and off-policy (off.) training methods are highlighted with distinct colors. **For each method, all reported scores are derived from the single checkpoint that achieved the highest average accuracy across the five benchmarks.** Best scores are **bolded** and marked with (1st), while second-best scores are *italicized* and marked with (2nd).
>
> |Method|AIME24|AIME25|Math-500|Minerva|Olympiad Bench|Avg.|
> |:---|:---:|:---:|:---:|:---:|:---:|:---:|
> |**Qwen2.5-32B-Base (800 training steps)**|||||||
> |GRPO (off.)|30.63|22.29|88.00|41.18|54.37|47.29|
> |GSPO (off.)|33.33|22.29|87.60|**48.53 (1st)**|55.56|49.46|
> |Clip-Higher (off.)|**38.33 (1st)**|**29.79 (1st)**|87.60|45.22|56.44|51.48|
> |KL-Cov (off.)|35.62|27.50|87.40|44.49|55.11|50.02|
> |80/20 (off.)|*38.12 (2nd)*|*28.75 (2nd)*|87.00|45.22|58.37|*51.49 (2nd)*|
> |Lp-Reg (off.)|37.71|24.58|**90.20 (1st)**|40.81|59.70|50.60|
> |GRPO (on.)|28.54|22.50|86.60|44.85|*60.30 (2nd)*|48.56|
> |GRPO+Entropy Loss (on.)|3.75|1.88|60.80|27.94|22.22|23.32|
> |80/20 (on.)|32.50|28.54|89.40|45.59|57.63|50.73|
> |Lp-Reg (on.)|*38.12 (2nd)*|27.08|*90.00 (2nd)*|*46.32 (2nd)*|**61.19 (1st)**|**52.54 (1st)**|
> |**Qwen3-14B-Base (1,000 training steps)**|||||||
> |GRPO (off.)|34.38|27.08|89.20|49.26|55.70|51.13|
> |GSPO (off.)|41.46|34.58|88.60|**50.74 (1st)**|59.85|55.05|
> |Clip-Higher (off.)|41.67|32.71|**95.00 (1st)**|47.43|64.00|56.16|
> |KL-Cov (off.)|*49.17 (2nd)*|*34.79 (2nd)*|93.00|47.43|62.07|57.29|
> |80/20 (off.)|43.96|34.58|91.80|48.16|60.89|55.88|
> |Lp-Reg (off.)|46.25|34.17|92.40|48.16|64.44|57.08|
> |GRPO (on.)|46.04|34.38|93.00|48.53|65.19|57.43|
> |GRPO+Entropy Loss (on.)|37.29|25.21|88.20|46.32|54.96|50.40|
> |80/20 (on.)|47.29|32.50|91.60|*50.37 (2nd)*|*65.78 (2nd)*|*57.51 (2nd)*|
> |Lp-Reg (on.)|**50.83 (1st)**|**37.92 (1st)**|*94.40 (2nd)*|49.26|**68.44 (1st)**|**60.17 (1st)**|

---

> ### Author Response · Authors · 2025-11-24
> **Response to Reviewer 2rNy (2/5)**
>
> > **Q-1.2 Because the reported improvement numbers themselves look small, it is unclear whether the differences are statistically significant.**
>
> A-1.2: We would like to clarify this concern via **(1)per-benchmark peak performance analysis**, **(2)pass@k evaluation**, and **(3)consistent improvement across model sizes, families, and domains**:
>
> **(1)Per-Benchmark Peak Performance Analysis**
>
> To address the concern that aggregated metrics might obscure the model's specific strengths and the improvement may look small, we provide a detailed per-benchmark peak performance analysis in **Appendix B.5** (Table 3). This table reports the maximum score achieved for each benchmark individually.
>
> As shown in **Table R1** (Table 3 in pdf) below, **Lp-Reg achieves the highest peak scores on four out of five benchmarks.** Even on Minerva, where it ranks second, the gap is marginal (trailing the best score by only $1.47$ percentage points on Qwen2.5-32B). Conversely, Lp-Reg achieves substantial gains on the most challenging tasks: it outperforms the second-best method by a relative margin of **10.78% on AIME24** (Qwen2.5-32B) and **10.77% on AIME25** (Qwen3-14B). These significant improvements on the hardest benchmarks underscore that the gains from Lp-Reg are not "small" but are statistically meaningful steps forward in complex reasoning.
>
> **Table R1 (Table 3 in pdf):** Per-benchmark peak performance on five mathematical reasoning benchmarks. **Note that the scores reported represent the maximum value achieved for each specific benchmark individually; thus, scores within a single row may originate from different training checkpoints.** Best scores are **bolded** while second-best scores are *italicized*. The relative accuracy improvement of Lp-Reg over the next best method is indicated in parentheses.
>
> |Methods|AIME24|AIME25|Math-500|Minerva|Olympiad Bench|
> |:---|:---:|:---:|:---:|:---:|:---:|
> |**Qwen2.5-32B-Base**||||||
> |GRPO (off.)|30.63|23.75|88.00|46.69|56.00|
> |GSPO (off.)|36.88|26.46|89.00|**49.63**|56.30|
> |Clip-Higher (off.)|39.58|**32.71**|88.80|*48.90*|58.22|
> |KL-Cov (off.)|36.88|29.38|89.00|48.16|56.89|
> |80/20 (off.)|*40.62*|30.21|*90.80*|48.16|58.81|
> |Lp-Reg (off.)|37.71|26.88|90.20|43.38|60.15|
> |GRPO (on.)|32.50|23.54|88.80|47.79|*60.30*|
> |GRPO+Entropy Loss (on.)|3.75|2.50|60.80|32.72|22.22|
> |80/20 (on.)|35.00|28.54|90.00|47.79|58.81|
> |Lp-Reg (on.)|**45.00**(+10.78%)|**32.71**(+0.00%)|**93.00**(+2.42%)|48.16(-2.96%)|**64.15**(+6.38%)|
> |**Qwen3-14B-Base**||||||
> |GRPO (off.)|35.83|27.71|91.00|48.16|59.56|
> |GSPO (off.)|43.75|*36.67*|91.60|50.74|61.04|
> |Clip-Higher (off.)|44.79|33.75|**95.00**|49.63|65.19|
> |KL-Cov (off.)|*49.38*|35.83|94.20|**51.84**|64.44|
> |80/20 (off.)|44.17|34.58|92.80|50.37|62.81|
> |Lp-Reg (off.)|48.75|34.79|94.40|49.63|*65.78*|
> |GRPO (on.)|46.04|35.42|93.80|50.37|65.63|
> |GRPO+Entropy Loss (on.)|37.29|28.54|90.60|48.53|57.93|
> |80/20 (on.)|47.29|35.00|94.00|50.37|*65.78*|
> |Lp-Reg (on.)|**51.88**(+5.06%)|**40.62**(+10.77%)|**95.00**(+0.00%)|*51.47*(-0.71%)|**70.37**(+6.98%)|

---

> ### Author Response · Authors · 2025-11-24
> **Response to Reviewer 2rNy (3/5)**
>
> ```
> continue to A-1.2
> ```
>
> **(2) Pass@k Evaluation**
>
> We further evaluate the exploration capability and robustness of our method by comparing the best pass@k rates. High Pass@k scores indicate that the model's exploration is diverse and effective, rather than lucky.
>
> As detailed in **Table R2** (Table 4 in pdf), Lp-Reg consistently achieves the highest Pass@k scores on both AIME24 and AIME25 across all model scales, often by a wide margin. For instance, on the Qwen3-14B model, Lp-Reg shows impressive gains on AIME25, achieving relative improvements ranging from **7.81% to 9.33%** compared to the baselines. On the Qwen2.5-32B model, it demonstrates a minimum relative improvement of **5.97%** on AIME24. These robust Pass@k results provide strong evidence that Lp-Reg effectively sustains meaningful exploration throughout long-horizon RLVR training, resulting in more diverse and successful reasoning rollouts.
>
> **Table R2 (Table 4 in pdf):** Per-benchmark peak pass@k results on the challenging AIME24 and AIME25 benchmarks. Similar to Table A-1, **scores reported denote the peak pass@k rate for each metric separately, implying they may be derived from different checkpoints.** Best scores are **bolded** and second-best scores are *italicized*. The relative improvement of Lp-Reg is indicated in parentheses.
>
> |Methods||AIME24|||AIME25||
> |:---|:---:|:---:|:---:|:---:|:---:|:---:|
> ||Pass@2|Pass@4|Pass@8|Pass@2|Pass@4|Pass@8|
> |**Qwen2.5-32B-Base**|||||||
> |GRPO (off.)|40.06|49.87|58.10|29.11|36.25|44.75|
> |GSPO (off.)|46.83|57.62|66.78|32.86|38.84|45.04|
> |Clip-Higher (off.)|48.11|57.80|68.32|*35.92*|*43.27*|*51.29*|
> |KL-Cov (off.)|46.89|55.94|64.61|35.44|41.60|49.39|
> |80/20 (off.)|48.97|56.52|64.29|34.08|41.35|49.47|
> |Lp-Reg (off.)|*49.69*|*59.75*|*69.21*|33.75|42.44|50.80|
> |GRPO (on.)|42.08|51.74|61.95|29.19|35.83|43.20|
> |GRPO+Entropy Loss (on.)|6.89|11.88|19.08|4.00|6.06|10.11|
> |80/20 (on.)|45.06|55.33|63.40|35.28|41.64|48.54|
> |Lp-Reg (on.)|**53.33**(+7.33%)|**63.50**(+6.28%)|**73.34**(+5.97%)|**38.28**(+6.57%)|**45.52**(+5.20%)|**53.12**(+3.57%)|
> |**Qwen3-14B-Base**|||||||
> |GRPO (off.)|45.31|54.81|64.09|34.14|41.00|48.29|
> |GSPO (off.)|54.11|63.67|71.05|*44.39*|*51.97*|*59.67*|
> |Clip-Higher (off.)|56.00|*66.85*|*74.91*|40.19|48.31|57.35|
> |KL-Cov (off.)|*59.47*|66.84|74.52|42.22|49.98|58.65|
> |80/20 (off.)|57.14|66.25|72.05|41.50|49.26|59.03|
> |Lp-Reg (off.)|58.08|64.23|71.41|40.86|46.30|52.39|
> |GRPO (on.)|55.19|63.93|70.48|42.86|49.90|57.85|
> |GRPO+Entropy Loss (on.)|47.44|57.53|66.34|34.86|41.62|48.09|
> |80/20 (on.)|56.97|63.66|71.66|42.28|49.76|57.39|
> |Lp-Reg (on.)|**62.67**(+5.38%)|**71.04**(+6.27%)|**79.85**(+6.59%)|**48.53**(+9.33%)|**56.03**(+7.81%)|**64.95**(+8.85%)|
>
> **(3) Consistent Improvement Across Model Sizes, Series, and Domains**
>
> Finally, regarding statistical significance, while running multiple random seeds for large-scale RLVR is computationally prohibitive, the significance of Lp-Reg is strongly evidenced by its **unwavering consistency across diverse settings** on a cumulative total of 300,000 GPU-Hours experiments. We extended our training and evaluation to cover additional model sizes (8B), distinct model families (Llama-3), and diverse domains (Code and Science).
> *   **Various Model Sizes:** It consistently outperforms baselines on 8B, 14B, and 32B models.
> *   **Various Model Series:** It achieves substantial improvements on Qwen2.5, Qwen3, and Llama-3 (further detailed in **A-1.3**).
> *   **Various Domains:** It surpasses other methods in Math, Code, and Science benchmarks (further detailed in **A-1.3**).
>
> The probability of a method achieving such consistent gains across all these independent variables purely by chance is negligible, confirming the statistical significance of our results.

---

> ### Author Response · Authors · 2025-11-24
> **Response to Reviewer 2rNy (4/5)**
>
> > **Q1.3: Concerns about the robustness of the results on other domains and model families.**
>
> A1.3: In our revision, we have added **Section B.6** (Generalization Across Architectures and Domains) to address this concern, extending our evaluation to the Llama-3 family and to science and code domains.
>
> **(1) Extension to Different Model Families (Llama-3)**
>
> The vanilla Llama-3 series is known to present significant challenges for reasoning RLVR training [1,2]. To test the architectural robustness of Lp-Reg, we conducted experiments on a mid-trained **Llama3-OctoThinker-8B**[3].
> As presented in **Table R3** (Table 5 in pdf) below, our method consistently outperforms baselines, consistent with the success on Qwen models. Specifically, in on-policy training, **Lp-Reg(on.) achieves an absolute gain of 2.88% over the second-best method (GRPO)**. In the off-policy setting, the advantage is even more pronounced, with Lp-Reg(off.) surpassing the nearest competitor by at least **3.62% absolute accuracy**. These results confirm that Lp-Reg's effectiveness is not specific to the Qwen family but generalizes well to other foundational architectures.
>
>
> **Table R3 (Table 5 in pdf):** Main results on five mathematical reasoning benchmarks on Llama3-OctoThinker-8B. On-policy (on.) and off-policy (off.) training methods are highlighted with distinct colors. **Benchmark scores correspond to the same checkpoint that achieves the highest average test set accuracy across the whole training.** Best scores are **bolded** while second-best scores are *italicized*.
>
> |Method|AIME24|AIME25|Math-500|Minerva|Olympiad Bench|Avg.|
> |:---|:---:|:---:|:---:|:---:|:---:|:---:|
> |**Llama3-OctoThinker-8B**|||||||
> |GRPO (off.)|4.38|4.58|60.00|26.47|25.93|24.27|
> |GSPO (off.)|4.58|2.50|58.80|29.41|25.33|24.13|
> |Clip-Higher (off.)|11.88|3.75|61.80|23.16|26.96|25.51|
> |KL-Cov (off.)|7.71|4.58|55.00|23.16|22.96|22.68|
> |80/20 (off.)|10.00|7.50|59.00|18.75|27.56|24.56|
> |Lp-Reg (off.) (ours)|9.58|8.33|68.80|27.21|31.70|29.13|
> |GRPO (on.)|*15.42*|*12.50*|*76.20*|*33.09*|*43.26*|*36.09*|
> |80/20 (on.)|11.67|4.17|73.60|27.21|37.48|30.82|
> |Lp-Reg (on.) (ours)|**18.33**|**16.88**|**79.00**|**35.29**|**45.33**|**38.97**|
>
>
> **(2) Extension to Different Domains (Science and Code)**
>
> To verify domain robustness, we trained and evaluated models on **Science** (biology, chemistry, physics) and **Code Generation** tasks using Qwen3-8B-Base.
> *   **Code Generation:** The model was trained on the AReaL-boba-2-RL-Code[4] dataset and evaluated on LiveCodeBench[5]. On the LiveCodeBench (LCB-v5/v6) benchmarks, Lp-Reg achieves the highest average scores in both on-policy and off-policy categories, demonstrating its applicability to logic-heavy coding tasks.
> *   **Science:** The model was trained on the SCP-116k[6]  dataset and evaluated dataset on GPQA-diamond[7]. On the challenging PhD-level GPQA-diamond benchmark, Lp-Reg(on.) and Lp-Reg(off.) surpass their respective baselines by 2.34% and 2.59% in absolute accuracy.
> As shown in **Table R4** (Table 6 in pdf) below, the consistent improvements across Math, Science, and Code domains demonstrate that Lp-Reg is a domain-agnostic regularization technique capable of enhancing reasoning in diverse contexts.
>
>
> **Table R4 (Table 6 in pdf):** Results on science and code domains on Qwen3-8B-Base. On-policy (on.) and off-policy (off.) training methods are highlighted with distinct colors. **Benchmark scores correspond to the same checkpoint that achieves the highest average test set accuracy across the whole training.** Best scores are **bolded** while second-best scores are *italicized*.
>
> |Methods|LCB-v5|LCB-v6|Avg. LCB|GPQA-diamond|
> |:---|:---:|:---:|:---:|:---:|
> |**Qwen3-8B-Base**|||||
> |GRPO (off.)|27.32|27.43|27.38|39.71|
> |GSPO (off.)|28.29|26.57|27.43|47.16|
> |Clip-Higher (off.)|27.10|27.57|27.34|48.61|
> |KL-Cov (off.)|28.74|27.43|28.09|49.18|
> |80/20 (off.)|26.57|27.64|27.11|45.90|
> |Lp-Reg (off.)|**29.57**|27.57|*28.57*|*51.77*|
> |GRPO (on.)|27.47|*27.86*|27.67|50.63|
> |80/20 (on.)|28.29|27.36|27.83|48.42|
> |Lp-Reg (on.)|*28.89*|**29.00**|**28.95**|**52.97**|
>
> [1] Gandhi et al. Cognitive behaviors that enable self-improving reasoners, or, four habits of highly effective stars.
>
> [2] Liu et al. Understanding r1-zero-like training: A critical perspective.
>
> [3] Wang et al. Octothinker: Mid-training incentivizes reinforcement learning scaling.
>
> [4] Fu et al. Areal: A large-scale asynchronous reinforcement learning system for language reasoning.
>
> [5] Jain et al. Livecodebench: Holistic and contamination free evaluation of large language models for code.
>
> [6] Lu et al. Scp-116k: A high-quality problem-solution dataset and a generalized pipeline for automated extraction in the higher education science domain.
>
> [7] Rein et al. GPQA: A graduate-level google-proof q&a benchmark.

---

> ### Author Response · Authors · 2025-11-24
> **Response to Reviewer 2rNy (5/5)**
>
> > **Q2 (W2): I'm also wondering whether the proposed technique would eventually show a performance collapse after 1,000 training steps. Reporting the point at which their method begins to collapse (if it does) could be useful and interesting.**
>
> A2: To further validate the training stability, we extended the training of the Qwen2.5-32B base model to **3,000 steps**, incurring a computational cost of **81,204 GPU-Hours** (approximately 53 days on 64 $\times$ NVIDIA H20 GPUs). We ensured a strictly continuous, end-to-end training process: we maintained fixed hyperparameters from step 1 to step 3,000, without the use of a reference model or any hyperparameter resets based on intermediate checkpoints. To facilitate reproducibility and further research, we will publicly release our code and the corresponding training scripts to the community.
>
> **The results, visualized in Figure 2 of the pdf, show no sign of performance collapse:** As shown in the right subfigure, Lp-Reg maintains a stable, upward trend in test set accuracy throughout the entire 3,000 steps. The entropy curve (middle subfigure) stabilizes within the ideal range of $(0.1, 0.4)$, effectively preventing both entropy collapse and entropy explosion. We observed a minor accuracy fluctuation around step 2,700. Our analysis (left subfigure) attributes this to the model hitting the maximum context length limit at 8,192 tokens. Specifically, since the model failed to extract an answer within the 8,192 token limit, it received a negative advantage due to truncation (note that we do not apply an explicit length penalty). The model successfully adapted to this signal by shortening its reasoning chains around step 2,900, causing accuracy to recover and continue rising.
>
> This stable, long-horizon training over 3,000 steps on a large 32B model demonstrates the method's significant potential for long-horizon RL scaling, which is an important objective for RLVR in reasoning LLMs.
>
> ---
> **Thank you again for your insightful feedback and constructive suggestions. In addition to the responses above, we have revised the paper accordingly. Please let us know if our responses address your concerns. We sincerely appreciate your time and effort in helping us enhance our work.**

---

### Author Response · Authors · 2025-11-24
**Summary of Revisions to the Manuscript**

We sincerely thank the reviewers for their time and thoughtful feedback on our paper. Based on their insightful comments and suggestions, we have carefully revised the manuscript, with all changes marked in blue. The major updates in the newly uploaded version are as follows:

- Figure 2: Added the training dynamics showcasing the stable training of our Lp-Reg over 3,000 steps, utilizing 81,204 GPU-hours (53 days on 64 * Nvidia H20 GPUs) on the Qwen2.5-32B base model.

- Section 6.3: Added statistical evidence demonstrating that meaningful exploratory tokens (e.g., wait) have higher average probabilities than noise tokens (e.g., cost) in the low-probability region of next-token prediction. (Figure 8)
- Appendix B.3: Included data-driven guidelines for selecting initial values for hyperparameters. (Figures 11, 12)
- Appendix B.5: Reported the per-benchmark peak performance across five mathematical reasoning benchmarks. (Tables 3, 4)
- Appendix B.6: Reported results on Llama3-OctoThinker-8B, as well as results in code and science domains. (Tables 5, 6)
- Appendix B.7: Added an analysis of the computational overhead of the proposed Lp-Reg algorithm. (Listing 1, 2; Table 7)
- Appendix C.1: Provided a theoretical analysis of low-probability tokens and high-entropy tokens. (Proposition 1; Figures 14, 15)
- Appendix C.2:  Conducted a trajectory-level analysis of meaningful exploratory tokens (e.g., wait) and noise tokens (e.g., cost). (Figures 16, 17)
- Appendix C.3: Added a discussion and comparison between our Lp-Reg method (targeted at training stability) and Lopti (targeted at training efficiency). (Figure 18)

---

### Author Response · Authors · 2025-12-03
**Summary of Rebuttal**

Dear Reviewers, ACs, SACs, and PCs,

We sincerely thank you and the reviewers for the time dedicated to evaluating our work. We are grateful for the recognition of our **novel insight** (Reviewer 9MFy), **principled and effective method** (Reviewers 2rNy, 9MFy, CCqW), and **strong empirical results** (Reviewers 2rNy, 9MFy, CCqW, 2YYP).

Since our initial submission in September, we have continuously expanded this work, growing the manuscript from **22 to 34 pages** supported by **over 300,000 cumulative GPU-hours** of experiments. These updates address the initial concerns regarding robustness, stability, efficiency, and theoretical analysis.

However, the recent OpenReview system issue has disrupted the interactive discussion phase, preventing us from effectively engaging with reviewers to clarify these major improvements. Given this constraint, **we decided to withdraw the paper** to incorporate these extensive revisions into a future submission.

Below, we summarize how the **revised version** addresses the initial concerns, serving as a record of the substantial improvements made during this period.

**Reviewer 2rNy (Rating 4)**
*The reviewer questioned improvements, robustness, and long-term stability.*
- **Rebuttal:** We provided comprehensive new evidence:
    1.  **Improvements:** Clarified our evaluation protocol. Lp-Reg ranks **1st or 2nd** in average accuracy (Table 2) and dominates peak scores on 4/5 benchmarks (e.g., **+10.78% on AIME24**, Table 3).
    2.  **Robustness:** Extended evaluation to **Llama-3**, **Code**, and **Science** domains (App. B.6). Lp-Reg outperforms baselines on Llama3-OctoThinker-8B (Table 5) and excels in LiveCodeBench/GPQA (Table 6) using **identical hyperparameters**.
    3.  **Stability:** Conducted a massive **3,000-step training run** (81,204 GPU-Hours) on Qwen2.5-32B (Fig. 2). Results show continuous improvement without collapse, proving the capability for long-horizon scaling.

**Reviewer 9MFy (Rating 4 -> 6)**
*The reviewer asked about hyperparameters, noise filtering, and computational overhead. Note: Score raised after initial response.*
- **Hyperparameters:** We added **data-driven guidelines** (App. B.3) deriving rational values for $\rho$ and $\kappa$ from baseline dynamics, removing guesswork. Sensitivity analysis (App. B.4) confirms robustness.
- **Noise Filtering:** We added statistical evidence (Section 6.3, Fig. 8) proving meaningful exploratory tokens (e.g., "wait") maintain **~2x higher probability** than noise tokens in the low-probability tail, justifying our filtering.
- **Computational Overhead:** Added runtime analysis (App. B.7). Empirical overhead is negligible (**0.3% $\sim$ 0.5%**).

**Reviewer CCqW (Rating 4)**
*The reviewer asked about hyperparameters, computational overhead, and comparison with concurrent work.*
- **Hyperparameters:** We showed that a **single hyperparameter set** generalizes effectively across Qwen, Llama-3, Math, Code, and Science domains (App. B.6), refuting expert dependence.
- **Computational Overhead:** Clarified that Lp-Reg uses efficient parallel operations, not full-vocabulary sorting per token. Overhead is **<0.5%** (App. B.7).
- **Comparison with Lopti:** Added detailed comparison (App. C.3). Lopti targets *data efficiency* while Lp-Reg targets *training stability*. Our 1,000-step head-to-head (Fig. 18) shows Lp-Reg sustains exploration where Lopti plateaus.
- **Trajectory Analysis:** Added decomposition (App. C.2). "Reasoning sparks" appear consistently in both positive and negative trajectories; preserving them in failed attempts is key to sustained exploration.

**Reviewer 2YYP (Rating 4)**
*The reviewer asked for differentiation from high-entropy methods, theoretical support, and hyperparameters.*
- **Differences from Entropy-based methods:** We distinguished our work via **(1) Empirical Results:** Lp-Reg significantly outperforms high-entropy baselines (Table 2); **(2) Analysis:** High-entropy methods miss "blind spots": exploratory tokens occurring in low-entropy regions (App. C.1).
- **Theoretical Analysis:** Added **Proposition 1** (App. C.1), theoretically proving that tokens targeted by high-entropy methods are a **subset** of those captured by Lp-Reg. This explains why we capture valuable signals that entropy methods overlook.
- **Hyperparameters:** Referenced sensitivity analysis (App. B.4) and provided guidelines (App. B.3) to identify good starting points from data.

We believe the **Revised Version**, with its significantly expanded scope and rigorous validation, constitutes a solid contribution to the field.

Best,
The Authors

---

### Note · Authors · 2026-01-04

I have read and agree with the venue's withdrawal policy on behalf of myself and my co-authors.